# A Framework for Ice Sheet - Ocean Coupling (FISOC) V1.1

Rupert Gladstone[1], Benjamin Galton-Fenzi[2], David Gwyther[3], Qin Zhou[4], Tore Hattermann[5,10], Chen Zhao[3], Lenneke Jong[2], Yuwei Xia[6], Xiaoran Guo[6], Konstantinos Petrakopoulos[8], Thomas Zwinger[9], Daniel Shapero[7], and John Moore[1,6]

[1]Arctic Centre, University of Lapland, Rovaniemi, Finland
[2]Australian Antarctic Division
[3]University of Tasmania, Hobart, Australia
[4]Akvaplan-niva AS, Tromsø, Norway
[5]Norwegian Polar Institute, Tromsø Norway
[10]Energy and Climate Group, Department of Physics and Technology, The Arctic University - University of Tromsø, Norway
[6]Beijing Normal University, China
[7]University of Washington, Seattle, US
[8]Center for Global Sea Level Change, New York University Abu Dhabi, United Arab Emirates
[9]CSC IT Center for Science, Espoo, Finland

**Correspondence:** Rupert Gladstone
(RupertGladstone1972@gmail.com)

**Abstract.**

A number of important questions concern processes at the margins of ice sheets where multiple components of the Earth System, most crucially ice sheets and oceans, interact. Such processes include thermodynamic interaction at the ice-ocean interface, the impact of melt water on ice shelf cavity circulation, the impact of basal melting of ice shelves on grounded ice dynamics, and ocean controls on iceberg calving. These include fundamentally coupled processes in which feedback mechanisms between ice and ocean play an important role. Some of these mechanisms have major implications for humanity, most notably the impact of retreating marine ice sheets on global sea level. In order to better quantify these mechanisms using computer models, feedbacks need to be incorporated into the modelling system. To achieve this, ocean and ice dynamic models must be coupled, allowing run time information sharing between components. We have developed a flexible coupling framework based on existing Earth System coupling technologies. The open-source Framework for Ice Sheet – Ocean Coupling (FISOC) provides a modular approach to coupling, facilitating switching between different ice dynamic and ocean components. FISOC allows fully synchronous coupling, in which both ice and ocean run on the same time step, or semi-synchronous coupling in which the ice dynamic model uses a longer time step. Multiple regridding options are available, and multiple methods for coupling the sub ice shelf cavity geometry. Thermodynamic coupling may also be activated. We present idealised simulations using FISOC with a Stokes flow ice dynamic model coupled to a regional ocean model. We demonstrate the modularity of FISOC by switching between two different regional ocean models and presenting outputs for both. We demonstrate conservation of mass and other verification steps during evolution of an idealised coupled ice - ocean system, both with and without grounding line movement.

# 1 Introduction

The Antarctic and Greenland ice sheets have the potential to provide the greatest contributions to global sea level rise on century timescales (Church et al., 2013; Moore et al., 2013), with the greatest uncertainty in projections being due to the Marine Ice Sheet Instability (MISI; Mercer, 1978; Schoof, 2007; Robel et al., 2019). Ice dynamic behaviour is strongly sensitive to ocean currents, in particular the transport of warmer waters across the continental shelf causing high basal melt rates under ice shelves (Hellmer et al., 2012; Thoma et al., 2015). For Antarctica's Pine Island Glacier, which is likely undergoing un-

stable retreat due to MISI, ocean induced basal melting has been established as a trigger for MISI through both observational evidence (Christianson et al., 2016) and model studies (Gladstone et al., 2012; De Rydt et al., 2014; Favier et al., 2014). While MISI is fundamenatally a geometrically controlled phenomenon, its onset and the resulting rate of ice mass loss are strongly dependent on tight coupling between ice dynamic behaviour and ocean processes. Importantly, ocean-driven basal melt rates respond to the evolving geometry of ice shelf cavities (Timmermann and Goeller, 2017; Mueller et al., 2018), and

the grounded-ice dynamic behaviour responds to the evolving basal melt rates through their impact on the buttressing force provided by ice shelves to the grounded ice. While most ice sheet model based studies use relatively simple parameterisations for calculating basal melt rates beneath ice shelves, recent studies have highlighted limitations of this approach (De Rydt and Gudmundsson, 2016; Favier et al., 2019). In particular, melt parameterisations as a function of depth or thermal driving do not impose conservation of heat in the system, and none of the parameterisations fully capture the impact of evolving ice geometry

on cavity circulation.

Several projects to couple ice sheet and ocean models are underway, and most (including the current study) will contribute to the Marine Ice Sheet – Ocean Model Intercomparison Project first phase (MISOMIP1) and its child projects: the Marine Ice Sheet Model Intercomparison Project third phase (MISMIP+); and the Ice Shelf Ocean Model Intercomparison Project second phase (ISOMIP+; Asay-Davis et al. (2016)).

Coupling projects take different approaches to handling the different timescales of ice and ocean processes. An ice sheet flowline model coupled to a five box ocean model allows large ensemble simulations to be carried out, but is limited in terms of implementation of physical processes (Gladstone et al., 2012). A temporally synchronous approach allows the cavity geometry to evolve on the ocean time step as a function of the melt rates calculated by the ocean model and the ice dynamics calculated by the ice model (Goldberg et al., 2018). Asynchronous approaches incorporate a longer time step for ice than ocean, and

sometimes involve coupling through file exchange and with restarts for the ocean model (Seroussi et al., 2017; De Rydt and Gudmundsson, 2016; Thoma et al., 2015).

Here, we present a new, flexible Framework for Ice Sheet – Ocean Coupling (FISOC; Section 2). FISOC allows runtime coupling in which ice and ocean components are compiled as runtime libraries and run through one executable. FISOC provides the user choice of synchronicity options. Adopting Earth System Modeling Framework terminology (ESMF; Section 2), we

refer to an ocean model coupled through FISOC as an "ocean component" and an ice sheet or ice dynamic model coupled through FISOC as an "ice component". We use FISOC to couple two different 3D ocean models to an ice dynamic model and

present idealised simulations demonstrating mass conservation and consistent grounding line behaviour (Section 3). FISOC is also currently being used to contribute to ISOMIP+ and MISOMIP1.

## 2 Methodology

FISOC is an open source coupling framework built using the ESMF (Hill et al., 2004; Collins et al., 2005). FISOC aims to provide seamless runtime coupling between an existing ice sheet model and an existing ocean model for application to Antarctic ice sheet - ocean systems. In its current form, FISOC assumes that the important ice sheet - ocean interactions occur at the underside of a floating ice shelf, and that the lower surface of the ice shelf can be projected on to the horizontal plane.

FISOC aims to provide flexibility and computational efficiency through the following key features:

– Flexible modular architecture (Section 2.1) facilitates swapping between different ice components or between different ocean components according to purpose (Section 2.2).

    – Access to ESMF tools allows multiple regridding and interpolation options, including between regular grids and unstructured meshes (Section 2.3).

    – Multiple options for handling differing ice and ocean time scales include fully synchronous coupling, passing rates of
change, time averaging of variables (Sections 2.4 and 2.5).

    – Flexible run-time control over the exchange of variables allows specific coupling modes to be (de)activated as required, e.g. geometric coupling, thermodynamic coupling.

    – Grounding line movement (Section 2.8) is implemented using geometry change rates and a modified wet/dry scheme in the ocean component, with multiple options available for updating cavity geometry (Section 2.5).

– Flexibility for parallelisation options. Currently sequential coupling is implemented, but any combination of sequential and concurrent parallelisation is possible with minimal coding effort (see also Section 2.1.1).

    – ESMF compatibility means that FISOC can be embedded within any ESMF-based modelling system, e.g. as a regional model within a global model.

    – ESMF compatibility also means that additional ESMF components (e.g. an atmosphere model) could easily be added to
the coupled system.

These features are described further in the following sections and in the FISOC manual, which can be found in the FISOC repository (see "code availability" at the end of this paper).

### 2.1 Software design

While coupled models in Earth System science have been in existence for decades, and such coupled models are often viewed
as single entities (ocean - atmosphere general circulation models for example), the field of coupled ice sheet - ocean modelling

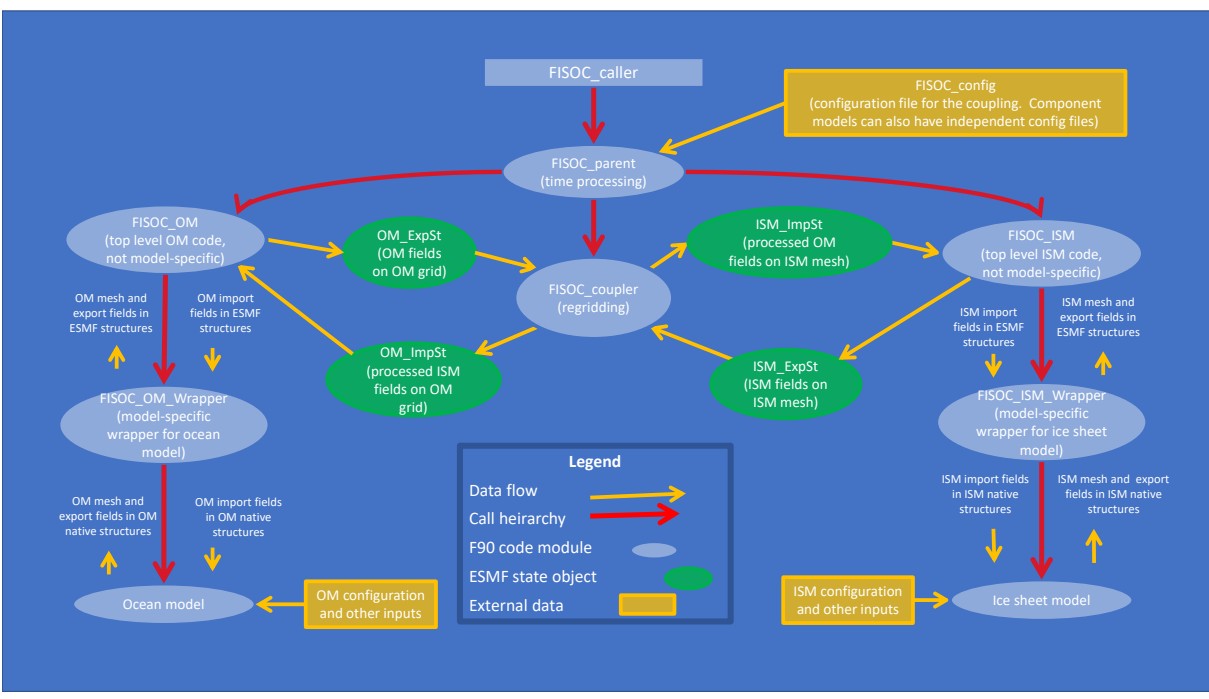

**Figure 1.** Overview of FISOC code structures. "OM" and "ISM" are short for Ocean and Ice Sheet Model (or component) respectively. "ImpSt" and "ExpSt" are short for Import and Export State respectively.

is relatively young. FISOC is intended as a framework for coupling independent models rather than as a coupled model in itself. Building and running a coupled ice sheet - ocean model is currently more complex than building and running both an ice and an ocean model independently. FISOC aims to minimise the additional complexity.

The ice and ocean components may use their standard run time input files, and their paths are set in a FISOC run time
configuration file, along with information about time stepping and variables to be exchanged.

FISOC adopts the hierarchical modular structure of the Earth System Modelling Framework. The FISOC code structures are summarised in Figure 1. A top level executable calls a FISOC parent module (this could in principal also be embedded within a larger coupled model framework). The parent module coordinates calling of the ice, ocean and regridding components. Regridding is one of the reasons to make use of ESMF, described further in Section 2.3. The ice and ocean components are
independent models, not included in the FISOC code repository, compiled as libraries to be called by FISOC at run time. On each side (ice and ocean) of the coupling is a model-specific wrapper, whose main run time functions are:

– Call the component's initialise, run and finalise routines as required.

– Convert the component's grid or mesh to ESMF format, using ESMF data structures.

- Read from, or write to, the component's required state variables, converting between the component's native data structures and ESMF data structures.

Further processing of variables (such as calculating rates of change) is implemented by the ice and ocean generic code modules.

Incorporating a new ice or ocean component into FISOC can be straightforward, depending on the existing level of ESMF compatibility of the new component. Models able to provide mesh information and variables in ESMF data structures can be very easily built in to FISOC. The only coding required for a new component is a new model-specific wrapper in the FISOC repository. Copying an existing wrapper can be a viable starting point.

### 2.1.1 Sequential parallelism

FISOC currently adopts a sequential parallelism paradigm. Each component runs on the full set of available Persistent Execution Threads (PETs). PET is an ESMF abstraction catering for multiple parallelism options. FISOC has so far used only the Message Passing Interface (MPI), in which one PET wraps one MPI process.

The sequential workflow is illustrated in Figure 2. The order of events during time stepping is as follows: The ocean component is called for the full number of ocean time steps required to complete one coupling interval. Ocean outputs are then regridded and passed to the ice component, which also runs for as many time steps as are required to complete one coupling interval. The ice component outputs are then regridded and passed to the ocean component. The ice component time step size is equal to the coupling interval for all simulations in the current study.

The initialisation is not shown in Figure 2, but we note that this is similar to the run time event order: the ocean component is initialised first, followed by regridding and then the ice component. There are two initialisation phases for each component, allowing for the possibility that variables may be needed to be passed from ice to ocean component in order to finalise initialisation.

This ordering of events imposes a lag in the sytem: While the ice component receives ocean variables for the current coupling interval, the ocean component only recieves ice variables for the previous coupling interval. This could be reversed (running the ice component before the ocean component) or could be modified such that both components receive variables from the other component for the previous coupling interval.

While FISOC implements sequential parallelism, ESMF also supports concurrent parallelism. Concurrent parallelism allows different components to run at the same time on different subsets of the available PETs. This approach is beneficial when different components have very different computational costs and parallel scaling: a cheap component that scales poorly is more effectively run on a subset of the available PETs, and concurrent parallelism allows this to be implemented more computationally efficiently than sequential. This could easily be implemented in FISOC if it becomes necessary, as the components, which utilise MPI, are assigned a distinct MPI communicator during initialisation. This communicator could be made to represent a subset of the available PETs. In principal concurrent parallelism also offers sub-time step coupling: it is possible to exchange variables between components during convergence of numerical schemes. Such coupling is unlikely to be implemented within

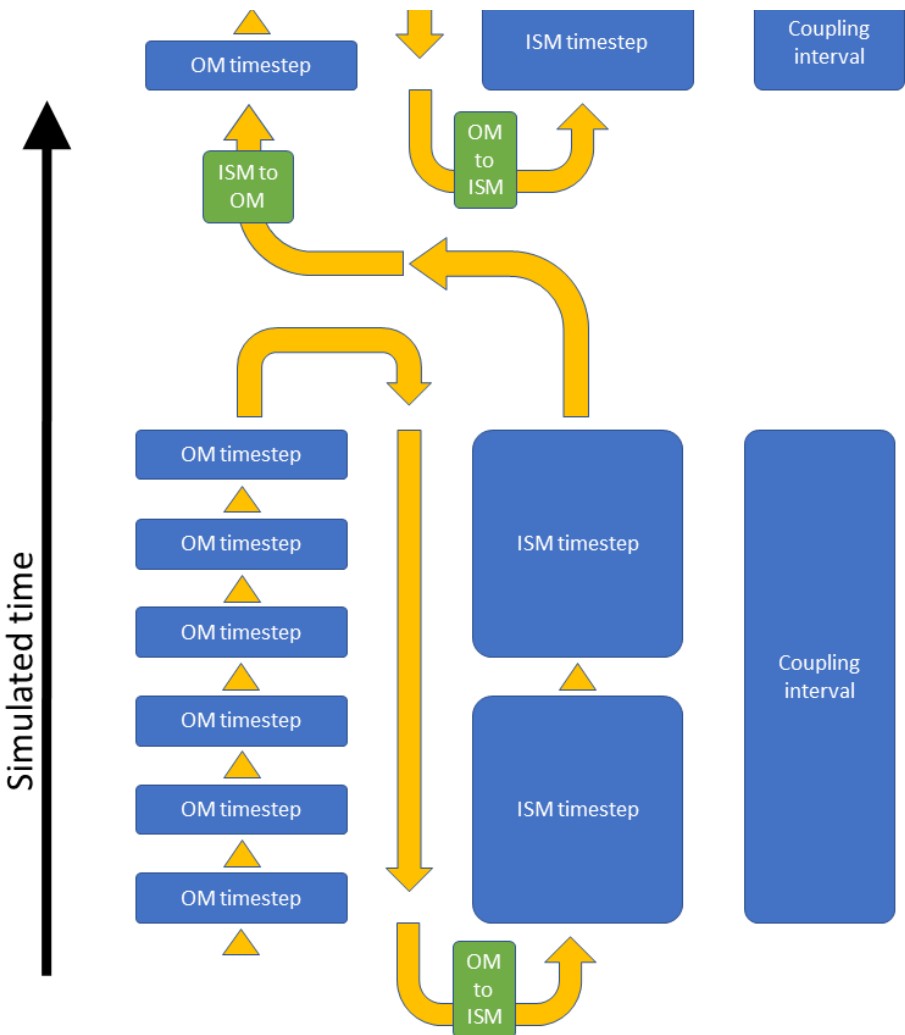

**Figure 2.** FISOC workflow. "OM" and "ISM" are short for Ocean and Ice Sheet Model respectively. The black arrow indicates the direction of simulated time. The yellow arrows indicate the order of events during a FISOC simulation. The green boxes indicate when regridding and passing of variables between components occurs. The length of the blue boxes in the vertical indicates example relative size of time steps and coupling interval (this is illustrative; in practice there will be many more OM time steps per ISM time step and the ISM time step size will usually equal the coupling interval).

FISOC as the timescales for ice and ocean components are so different. While sequential coupling imposes a lag between components (described above), concurrent coupling implemented in FISOC would impose a lag in both components: exchange of variables in both directions would occur at the end of the coupling interval.

**Table 1.** Ice and ocean components currently coupled through FISOC.

| Type | Name | Notes |
|------|------|-------|
| OM | ROMS | 3D, gridded, sigma coord |
| OM | FVCOM | 3D, unstructured mesh, sigma coord |
| ISM | Elmer/Ice | 3D, full Stokes and shallow models |

### 2.1.2  Error handling

The ESMF adopts a defensive strategy to error handling: All errors are logged and passed back up the call stack. The calling routine has the option of attempting to continue running in the event of errors occurring. As the call structure between FISOC and ESMF is one-way (FISOC routines may call ESMF routines but not vice versa), all such errors are eventually returned to FISOC.

FISOC adopts a fail-fast approach. Errors are generally considered to be fatal, in which case FISOC will log error information and finalise both ice and ocean components and ESMF. FISOC also aims to provide consistency checks, most of which are considered fatal if not passed. For example, ice and ocean input files might both contain time stepping information, potentially duplicating information in the FISOC run-time configuration file, and these can be checked for consistency in the model-specific wrappers. The general intention is to stop running if something unexpected happens and provide a meaningful message to the user about why.

There are a few cases where ESMF errors can be handled at run time. Details can be found in the FISOC manual, which can be accessed from the FISOC repository (see "code availability" at the end of this paper).

## 2.2  Components

FISOC is designed to facilitate swapping between different ocean or ice components. Currently two different ocean components and one ice component are available through FISOC. Table 1 summarises components currently coupled into FISOC. In some cases, a non-standard build of the component is required for FISOC compatibility, and these are described in the FISOC manual, which can be obtained through the FISOC repository (Section 2.1).

The ice component Elmer/Ice (Gagliardini et al., 2013) is a powerful, flexible, state-of-the-art ice dynamic model.

The Regional Ocean Modeling System (ROMS; Shchepetkin and McWilliams, 2005) is a 3D terrain-following, sigma-coordinate ocean model that has already been adapted to use in ice shelf cavities (Galton-Fenzi et al., 2012). The module for ice shelf cavities implemented in the Finite Volume Community Ocean Model (FVCOM; Chen et al., 2003) provides non-hydrostatic options, a horizontally unstructured mesh that lends itself to refinement, and may be more suited to small scale processes such as ice shelf channels (Zhou and Hattermann, 2020).

## 2.3 Regridding

As stated above, FISOC provides coupling on a horizontal plane onto which the lower surface of an ice shelf can be projected. It is this plane on which ice and ocean properties are exchanged through the FISOC framework. Adapting the FISOC code to handle a vertical ice cliff is expected to be straightforward, and would be desirable for application to the Greenland ice sheet. More complex 3D ice-ocean interface geometries are challenging not only for FISOC but also for the current generation of ice sheet and ocean models.

FISOC has access to all the run-time regridding options provided by ESMF. These include nearest neighbour options, conservative options, patch recovery and bilinear regridding. These options are available for structured grids and unstructured meshes. FISOC requires that both ice and ocean components define their grid or mesh on the same coordinate system, and that both components use the same projection. All FISOC simulations to date have used a Cartesian coordinate system (i.e. all components have so far used Cartesian coordinates).

Our current FISOC setup does not meet the requirements for all forms of ESMF regridding. Specifically, the conservative methods, when an unstructured mesh is involved, require that field values are defined on elements and not on nodes. Elmer, by default, provides field values on nodes, but can also provide element-wide values or values on integration points within elements. We will need to either map nodal values to element values or utilise element-type variables in order to use conservative regridding, and this is intended as a future development.

When using FISOC to couple Elmer/Ice to ROMS, the ROMS grid extends beyond the Elmer/Ice mesh. This is due to ROMS using a staggered grid (Arakawa C-grid) and ghost cells extending beyond the active domain. This necessitates the use of extrapolation. ESMF regridding methods provide options for extrapolation, which are used here. Simulations in the current study use either nearest "Source TO Destination" (STOD, a form of nearest neighbour) regridding or use bilinear interpolation (in which case nearest STOD is used only for destination points that lie outside the source domain).

We use subscripts with square brackets, $_{[X]}$, where $X$ is either $O$ (ocean component) or $I$ (ice component), to denote a variable that exists in both ice and ocean components with the same physical meaning, but potentially different values due to being represented on different grids/meshes.

## 2.4 Coupling timescales

The timescales for sub-shelf cavity circulation behaviour are in general much shorter than the timescales for ice flow and geometry evolution (typically minutes to days instead of years to centuries). Typical time step sizes are correspondingly smaller for ocean models (seconds to minutes) than for ice sheet models (days to months). A single ice sheet model time step, if the Stokes equations are solved in full, will typically require orders of magnitude more computational time than a single ocean time step. Due to the combination of these two reasons the ice and ocean components of FISOC will in general use different time steps, with the ice time step size being much larger. We define relevant terminology for coupling timescales:

– **Fully synchronous coupling.** The ice and ocean components have the same time step size, and exchange variables every time step.

- **Semi synchronous coupling.** The ice component has a larger time step than the ocean component, but the ocean component's cavity geometry and grounding line position are allowed to evolve on the ocean time step (e.g. by using ice velocities from a previous ice time step or rates of change based on the most recent two time steps).

- **Asynchronous coupling.** The ice component has a larger time step than the ocean component. Cavity geometry is updated on the ice component time step or less frequently.

- **Coupling interval.** The time interval at which the ice and ocean components exchange variables.

In the current study, FISOC sets the coupling interval equal to the ice component time step size. This is an exact multiple of the ocean model time step size. More generally (for potential future experiments), FISOC calls each component for a fixed time period and allows the component to determine its own time stepping within that period. In principal, adaptive time stepping could be implemented within this framework, so long as each component runs for the required amount of simulated time. FISOC does not currently provide an option to vary the coupling interval during a simulation, but this could be implemented if needed.

FISOC is flexible with regard to time processing of ocean or ice variables. It is possible to cumulate variables, calculate averages, or use snapshots. In the current study, the ocean components (both ROMS and FVCOM) calculate averaged basal melt rates over the coupling interval and pass these averages through FISOC to the ice component. In the current study, as the ice component time step size is equal to the coupling interval for all simulations, no time processing of ice component variables is needed.

In principal, FISOC supports all three synchronicity options, though fully synchronous coupling is not practical to achieve when solving the Stokes equations for the ice component. The experiments carried out for this paper use semi synchronous coupling with cavity geometry evolution as described in Section 2.5.

Goldberg et al. (2018) and Snow et al. (2017) implement fully synchronous coupling, whereas Seroussi et al. (2017) and Favier et al. (2019) implement asynchronous coupling with ocean restarts.

## 2.5 Handling cavity evolution

The evolution of cavity geometry under the ice shelf, defined by a reference ice draft, $z_d$ (positive upward), and grounding line location, is calculated by the ice component forced by the melt rates passed from the ocean component. We refer to $z_d$ as a "reference" ice draft because the ocean component may further modify the ice draft according to the dynamic pressure field. The ocean component's "free surface" variable, $\zeta$, represents, for the open ocean, the height of the upper surface of the ocean domain relative to a mean sea level. Under the ice shelf, $\zeta$ represents the deviation of the upper surface of the ocean domain relative to the reference ice draft $z_d$ (similar to Goldberg et al. (2018)). To summarise the meaning of the key variables: $z_{d[I]}$ is the reference ice draft computed by the ice component; $z_{d[O]}$ is the same but regridded for the ocean component; $(z_{d[O]} + \zeta)$ is the actual ice draft according to the ocean component.

Given the potential for non-synchronicity of the ice and ocean component time stepping, several methods are implemented in FISOC for the ocean to update its representation of $z_d$. All the processing options described below are applied on the ocean grid after the ice component representation of ice geometry has been regridded (i.e. $z_{d[I]}$ regridded to $z_{d[O]}$).

**Most recent ice**. The simplest option is that the ocean component uses the ice draft directly from the most recent ice component time step. If fully synchronous coupling is used, this option should be chosen. The main disadvantage of this approach for semi or asynchronous coupling is that, due to the much longer time step of the ice component, the ocean component will experience large, occasional changes in ice draft instead of smoothly evolving ice draft. This could be both physically unrealistic and potentially numerically challenging for the ocean component.

**Rate**. The vertical rate of change of ice draft, $\frac{dz_d}{dt}$, is calculated by FISOC after each ice component time step using the two most recent ice component time steps. If we assume that the ice component completes a time step at time $t$, the rate at this time is given by

$$\frac{dz_{d[O,t]}}{dt} = \frac{z_{d[I,t]} - z_{d[I,t-\Delta t_I]}}{\Delta t_I} \tag{1}$$

where $z_{d[O,t]}$ is the ocean component's reference ice draft at time $t$, $z_{d[I,t]}$ is the ice component's reference ice draft at time $t$, $z_{d[I,t-\Delta t_I]}$ is the ice component's reference ice draft at time $t - \Delta t_I$, and $\Delta t_I$ is the ice component time step size. This rate of change is used by the ocean component to update the cavity geometry until the next ice component time step completes. In this sense the ocean component lags the ice component as mentioned in Section 2.1.1. This approach provides temporally smooth changes to the ocean representation of the ice draft, but has the potential for the ice and ocean representations to diverge over time as a result of regridding artefacts.

**Corrected rate**. The same as above, except that a drift correction is applied to ensure ice and ocean representations of cavity geometry do not diverge.

$$\frac{dz_{d[O,t]}}{dt} = \frac{z_{d[I,t]} - z_{d[I,t-\Delta t_I]} + f_{cav}\left(z_{d[I,t]} - z_{d[O,t]}\right)}{\Delta t_I} \tag{2}$$

where $f_{cav}$ is a cavity correction factor between 0 and 1. Equation 2 is applied at coupling time steps, and the calculated rate of cavity change is then held constant during ocean component evolution until the coupling interval completes. Conceptually, this option prioritises ice - ocean geometry consistency over mass conservation.

**Linear interpolation**. The ocean representation of the ice draft is given by temporal linear interpolation between the two most recent ice sheet time steps. This imposes additional lag of the ocean component behind the ice component.

The above options are all implemented in FISOC, but only the "rate" and "corrected rate" approaches are used in the current study.

The cavity geometry may be initialised independently by ice and ocean components. In this case, the user must ensure consistency. It is also possible for the cavity geometry from the ice component to be imposed on the ocean component during FISOC initialisation. This ensures consistency.

Handling cavity evolution is a little more complicated in the case of an evolving grounding line, as discussed in Section 2.8 below.

## 2.6 Thermodynamics at the ice-ocean interface

Exchange of heat at the ice-ocean interface is handled within the ocean model. Like many ocean models, FVCOM and ROMS adopt the three-equation formulation for thermodynamic exchange (Hellmer and Olbers, 1989; Holland and Jenkins, 1999; Jenkins et al., 2010). This parameterisation assumes that the interface is at the in situ pressure freezing point, and that there is a heat balance and salt balance at the interface. Both ROMS and FVCOM assume constant turbulent transfer coefficients for scaling the heat and salt fluxes through the interface, with thermal and saline exchange velocities calculated as the product of these coefficients with friction velocity. Further details of the ROMS and FVCOM specific implementations of the three-equation formulation are given by Galton-Fenzi et al. (2012) and Zhou and Hattermann (2020) respectively. An ablation or melt rate is calculated for each ocean model grid cell, which is then passed to FISOC as a boundary condition for the lower surface of the ice model at the coupling time interval.

Internally, both ocean models account for the thermodynamic effect of basal melting by imposing virtual heat and salt fluxes within a fixed geometry at each ocean model time step, to mimic the effects of basal melting, rather than employing an explicit volume flux at the ice-ocean interface. Independent of this, a geometry change is passed back from the ice model through FISOC at after each coupling interval (including the effect of melting/freezing, as well as any ice dynamical response), which is used to update the ocean component cavity shape (Section 2.5).

For some applications, conductive heat fluxes into the ice shelf due to vertical temperature gradients in the ice at the ice-ocean interface are required by the three-equation parameterisation to calculate the flux balance at the ice ocean interface. While ice-ocean thermodynamic parameterisations in ocean-only models must make an assumption about this temperature gradient, FISOC can pass the temperature gradient from the ice component directly to the ocean component. This feature is not demonstrated in the current study, but will be properly tested in future studies.

Non-zero basal melt rates may be calculated by the ocean component in regions that are defined as grounded by the ice model. This could occur due to isolated patches of ungrounding upstream of the grounding line or to discrepancies between the ice and ocean component's representation of the grounded region. Basal melt rates are masked using the ice component's grounded mask before being applied within the ice component. This has the potential to impact on mass conservation in the coupled system. Future studies utilising conservative regridding will ensure that passing masked field variables between components remains conservative.

## 2.7 Interface pressure

Aside from the geometry evolution, an ocean boundary condition for pressure at the ice-ocean interface, $P_{interface}$, must be provided to the ocean component. FISOC can pass pressure directly from ice to ocean components. However, using actual ice overburden directly as an upper ocean boundary condition results in higher horizontal pressure gradients at the grounding line (and for dry cells, see Section 2.8) than ocean models can typically handle (Goldberg et al., 2018). In the current study, the ocean component uses the reference ice draft (see Section 2.5) to estimate a floation pressure. ROMS assumes a constant

reference ocean density:

$$P_{interface} = -g\rho_{or}z_{d[O]} \tag{3}$$

where $g$ is acceleration due to gravity, $\rho_{or}$ is a reference ocean density and $z_{d[O]}$ is the ocean representation of ice draft (positive upward). For the current study, all simulations with ROMS use $\rho_{or} = 1027\,\mathrm{kg\,m^{-3}}$. FVCOM assumes a constant vertical ocean density gradient following Dinniman et al. (2007):

$$P_{interface} = -g(\rho_{o1} + 0.5\frac{d\rho_o}{dz}z_{d[O]})z_{d[O]} \tag{4}$$

where $\rho_o$ is ocean water density, $\rho_{o1}$ is ocean water density of the top ocean layer and the vertical ocean water density gradient,
$\frac{d\rho_o}{dz}$, is set to $8.3927 \times 10^{-4}\mathrm{kg\,m^{-4}}$.

## 2.8 Grounding line evolution

Grounding line movement in FISOC requires that both ice and ocean components support it. Numerical convergence issues place constraints in terms of mesh resolution for representing grounding line movement in ice sheet models (Vieli and Payne, 2005; Pattyn et al., 2006; Gladstone et al., 2010a, b; Cornford et al., 2013; Gladstone et al., 2017). While FISOC allows ice
draft to be passed to the ocean component (Section 2.5), FISOC does not impose the ice component grounding line position on the ocean component. Instead, the ocean component uses the evolving cavity geometry to evolve the grounding line.

A recent ice-ocean coupling study (Goldberg et al., 2018) used a "thin film" approach to allow grounding line movement. A thin passive water layer is allowed to exist under the grounded ice, and an activation criterion is imposed to allow the layer to inflate to represent grounding line retreat. The current study takes a conceptually similar approach, modifying the existing
wetting and drying schemes independently in both ROMS (Warner et al., 2013) and FVCOM. "Dry" cells are used for the passive water column under grounded ice and "wet" cells are used for the active water column under floating ice or the open ocean. The wet - dry mask is two dimensional, so while it is conventional to talk about dry or wet cells, this actually refers to dry or wet columns. The grounding line evolves in the two horizontal dimensions, and is represented in the ocean component as the vertical surface between dry and wet columns.
The original criterion in both ROMS and FVCOM for a cell to remain dry is given by:

$$\zeta - z_b < D_{crit} \tag{5}$$

where $z_b$ is the bottom boundary depth (bathymetry, or bedrock depth, positive upward), and $D_{crit}$ is a critical water column thickness for wet/dry activation. $D_{crit}$ is a parameter to be set by the user (typical values lie between 1 to 20m). Thus, cells with a water column thickness less than $D_{crit}$ are designated dry. Flux of water into dry cells is allowed, but flux of water out
of dry cells is prevented.

The FVCOM criterion for an element to be dry has been modified for the presence of a marine ice sheet/shelf system as follows:

$$-z_b + z_{d[O]} < D_{crit} \tag{6}$$

This is a purely geometric criterion based entirely on the geometry determined by the ice component. The ROMS criterion for a cell to be dry has been modified for the presence of a marine ice sheet/shelf system as follows:

$$\zeta - z_b - (z_{s[O]} - z_{d[O]} - D_{crit})\frac{\rho_i}{\rho_{or}} \leq 0 \tag{7}$$

where $z_{s[O]}$ is the ocean representation of ice sheet/shelf upper surface height. $z_{s[O]}$ is needed in this equation because the floatation assumption cannot be made for grounded ice. This equation essentially compares $\zeta$ against the height above buoyancy of the grounded ice. In other words, if the dynamic variations in ocean pressure are sufficient to overcome the higher ice pressure due to the positive height above buoyancy, the cell can become ungrounded. The conceptual difference between the FVCOM and ROMS wetting criteria is that ROMS allows dynamic ocean pressure variations to make minor grounding line adjustments relative to the grounding line determined by the ice geometry, whereas FVCOM uses just the ice geometry to determine grounding line position.

FISOC allows the ice component to pass any geometry variables to the ocean, such as ice draft, ice thickness, upper surface elevation, or rates of change of any of these. In the event that geometry variables other than $z_d$ are passed to the ocean, the same processing method is used as for $z_d$, as described in Section 2.5. In the current study, $\frac{dz_d}{dt}$ is passed to the ocean component, and in one case both $\frac{dz_d}{dt}$ and $\frac{dz_s}{dt}$ are passed (details in Section 3). When $\frac{dz_s}{dt}$ is passed, $\frac{dz_s}{dt}$ is processed the same way as $\frac{dz_d}{dt}$. If the grounding line problem is solved, and if $z_d$ is processed for passing to the ocean using the **Corrected rate** method, equation 2 is modified to account for the dry water column thickness, which is initialised to $D_{crit}$. The correction term changes from $f_{cav}\left(z_{d[I,t]} - z_{d[O,t]}\right)$ to $f_{cav}\left(\max(z_{d[I,t]}\ ,\ z_b + D_{crit}) - z_{d[O,t]}\right)$.

There are no connectivity restrictions on wetting and drying in either of the ocean components in the current study. This means that it is possible for individual cells, or regions containing multiple cells, that are upstream of the grounding line to become wet (i.e. to unground). This occurs on small spatial and temporal scales in ROMS (individual cells a short distance upstream of the grounding line sometimes become temporarily wet) but not at all in FVCOM (likely due to choice of wetting criterion).

## 3 Verification experiment design

Simulations are carried out on idealised domains as a proof of concept to demonstrate the coupling rather than to address scientific questions. Verification experiment 1 (VE1) aims to assess whether the coupled system conserves mass. Verification experiment 2 (VE2) aims to assess whether the ocean and ice representations of grounding line evolution are consistent.

### 3.1 Verification experiment 1: Floating adjustment

Verification experiment 1 (VE1) is a simple experiment in which a linearly sloping ice shelf is allowed to adjust toward steady state. The experiment is not run long enough to attain steady state, but enough to demonstrate evolution of the coupled system. See Table 2 for run length and a summary of other model choices and parameter values used in VE1.

**Table 2.** Model choices and input parameters used in verification experiment 1 (VE1, Section 3.1) and verification experiment 2 (VE2, Section 3.2) comprising four simulations in total: VE1_ER, VE1_EF, VE2_ER and VE2_EF. Component abbreviations in these simulation names are E (Elmer/Ice), R (ROMS), and F (FVCOM). "Semi-structured" refers to a mesh that is in principal unstructured, but in practice structure can be seen (See Figure 3 middle and lower panes).

| Choice or input | VE1_ER | VE1_EF | VE2_ER | VE2_EF |
|---|---|---|---|---|
| Ice component | Elmer/Ice | Elmer/Ice | Elmer/Ice | Elmer/Ice |
| Ocean component | ROMS | FVCOM | ROMS | FVCOM |
| Ice mesh | Unstructured | Semi-structured | Unstructured | Semi-structured |
| Ocean mesh or grid | Structured, staggered | Semi-structured | Structured, staggered | Semi-structured |
| Domain size | 30 km × 100 km | 31 km × 99 km | 30 km × 100 km | 31 km × 99 km |
| Regrid method | Bilinear | Nearest STOD | Bilinear | Nearest STOD |
| Ocean time step size | 200 s | 20 s | 100 s | 20 s |
| Ice time step size | 10 days | 10 days | 10 days | 10 days |
| Coupling interval | 10 days | 10 days | 10 days | 10 days |
| Run length | 100 a | 47 a | 46 a | 40 a |
| Cavity update method | **Rate** | **Rate** | **Corrected rate** | **Rate** |
| Cavity correction factor, $f_{cav}$ | n/a | n/a | 0.01 | n/a |
| Minimum water column $D_{crit}$ | n/a | n/a | 5m | 5m |
| Ocean density $\rho_{or}$ | $1027\,\mathrm{kg\,m^{-3}}$ | n/a | $1027\,\mathrm{kg\,m^{-3}}$ | n/a |
| Ice density $\rho_i$ | $910\,\mathrm{kg\,m^{-3}}$ | $910\,\mathrm{kg\,m^{-3}}$ | $910\,\mathrm{kg\,m^{-3}}$ | $910\,\mathrm{kg\,m^{-3}}$ |
| Ice temperature | $-5\,°\mathrm{C}$ | $-5\,°\mathrm{C}$ | $-5\,°\mathrm{C}$ | $-5\,°\mathrm{C}$ |

All ice and ocean vertical side boundaries are closed: There is no flow in or out of the domain. There is mass exchange
between the ice and ocean (and therefore also heat exchange). The coupling centers on the evolution of ice geometry: the ocean component passes an ice shelf basal melt rate to the ice component and the ice component passes a rate of change of ice draft to the ocean component.

We expect adjustment toward a uniform-thickness ice shelf to occur by two mechanisms:

1. Ice dynamics. The gravitational driving force will tend to cause flow from thicker to thinner regions.

2. Melt/freeze. The greater pressures at greater depth should result in higher melt rates, with the potential for refreezing under thinner regions.

### 3.1.1 Domain size and meshes

The domain is 30 km across the expected direction of ice flow ($y$ direction) by 100 km along the flow ($x$ direction) for simulation VE1_ER. However, ocean component FVCOM (used in VE1_EF) uses a semi-structured (in principal unstructured but in practice exhibiting some structure) mesh with dimensions 31 km by 99 km. This results from an auto-generated-mesh method using a uniform resolution of 2 km for its triangular elements. FISOC does not in general require that ice and ocean component domains precisely overlap. Indeed the region of overlap is allowed to be small relative to the domains (for example an Antarctic ice stream interacts with the ocean only in its floating shelf, and the majority of the catchment may be grounded with no possibility to interact with the ocean for the duration of an intended simulation). However, given that we aim to address mass conservation in the coupled system, we choose to require precise domain match between ice and ocean components for the current study. Therefore, for simulations presented in the current study, the ice component has a slightly different domain when coupled to ROMS as compared to when coupled to FVCOM. For VE1_EF the ice component runs on an almost identical mesh to the ocean component. The only difference is at two diametrically opposite corners, where FVCOM prefers to maintain element shape but Elmer/Ice prefers to maintain a strictly rectangular domain (in order to facilitate imposition of consistent boundary conditions at the corners of the domain). These mesh differences are visually summarised in Figure 3.

### 3.1.2 Ice component setup

The initial geometry is of an ice shelf at floatation (i.e. hydrostatic equilibrium). The initial ice draft is given in m by

$$z_d = -450 + 400 \left( \frac{x}{100000} \right), \tag{8}$$

where $x$ is distance in m along the domain. The initial geometry does not vary across the ice flow ($y$ direction). Ice and ocean water densities used in the ice component are $\rho_i = 910 \ \mathrm{kg \, m^{-3}}$ and $\rho_{or} = 1027 \ \mathrm{kg \, m^{-3}}$ respectively. These densities, along with the floatation assumption, determine the ice upper surface.

The pressure acting on the underside of the ice shelf is given by Equation 3.

Temperature in the ice component is constant through space and time at $-5 \ °\mathrm{C}$.

VE1 includes ice flow and geometry evolution solving the Stokes equations directly. Glen's power law rheology with $n = 3$ is implemented (Glen, 1952; Gagliardini et al., 2013).

Zero accumulation is prescribed at the upper ice surface. The melt rate from the ocean component is applied at the lower surface. Flow through the vertical side boundaries is not allowed.

**Elmer/Ice specific details.** The Stokes equations are solved within Elmer/Ice (Gagliardini et al., 2013). A 2D horizontal mesh of triangles with an approximate element size of 1km (VE1_ER) or 2km (VE1_EF) is extruded in the vertical to give 11 equally spaced terrain-following layers with the bulk element shape being triangular prisms.

Ice model mesh for ER simulations

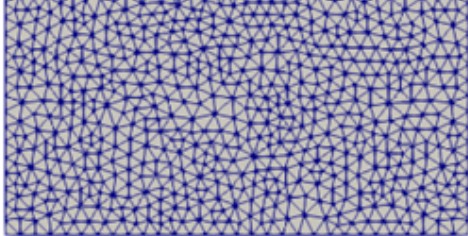

Ice model mesh for EF simulations

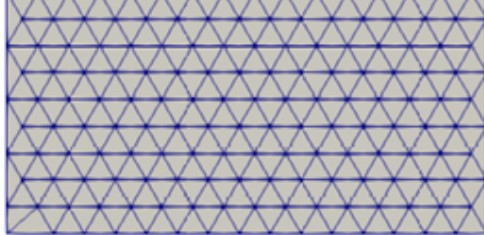

Ocean model mesh for EF simulations

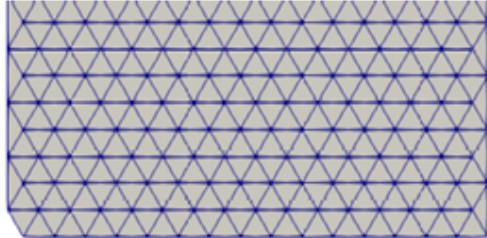

**Figure 3.** Unstructured meshes used in the current study. The first 15 km are shown. The ocean model in the ER simulations uses a structured grid.

### 3.1.3 Ocean component setup

The ocean bathymetry is set to $500$ m throughout the domain. The wet/dry scheme (Section 2.8) is not used in this experiment, as the whole domain is ice shelf cavity with no grounded ice. Boundaries are closed and rotation is disabled. Ocean potential

temperature is initialised at $-1.85\,°\text{C}$ and salinity at $34.6$ on the practical salinity scale. Ice-ocean thermodynamics are captured
by means of the three-equation parameterisation (Section 2.6).

The ocean conditions are chosen to represent a cold cavity ice shelf, such as the Amery Ice Shelf. In this configuration, both
basal melting and refreezing can occur.

**ROMS specific details.**

The horizontal resolution is a constant $1\,\text{km}$. There are 11 vertical layers, with a sigmoidal terrain-following distribution
configured to provide increased resolution near the top and bottom surfaces. The ROMS baroclinic (slow) time step size is
200 seconds, and there are 30 barotropic (fast) time steps for every slow time step. Interior mixing is parameterised with the
K-Profile Parameterisation (Large et al., 1994). Background vertical mixing coefficients for tracers and momentum are set
to constant values of $5.0e-5\,\text{m}^2\text{s}^{-1}$ and $1.0e-3\,\text{m}^2\text{s}^{-1}$, respectively, while horizontal viscosity and diffusivity are set to
$6.0\,\text{m}^2\text{s}^{-1}$ and $1.0\,\text{m}^2\text{s}^{-1}$ respectively.

**FVCOM specific details.** The horizontal grid resolution is $2\,\text{km}$ (defined by the distance between adjacent nodes within
a uniform triangular grid) and there are 11 uniformly spaced vertical terrain-following layers. Interior vertical mixing is pa-
rameterized using the Mellor and Yamada level 2.5 (Mellor and Yamada, 1982) turbulent closure model (vertical Prandtl
Number = 0.1) together with a constant background viscosity and diffusivity of $10^{-6}\,\text{m}^2\text{s}^{-1}$. An eddy closure parameteri-
sation (Smagorinsky, 1963) is used for the horizontal mixing of momentum (viscosity) and tracers (diffusivity) with both the
scaling factor and the Prandtl Number being 0.1. Both the barotropic time step and the baroclinic time step sizes are 20 seconds.

### 3.1.4 Coupling

The coupling interval is 10 days, the same as the ice component time step size. Cavity update method is **Rate** (Section 2.5).
For VE1_ER, the regridding method is bilinear with nearest STOD extrapolation for ocean cells that lie outside the ice domain
due to grid stagger. For VE1_EF, nearest STOD regridding is used, which results in a one to one mapping between ice and
ocean nodes due to the meshes being nearly identical (Section 3.1.1). There is no grounding line in this experiment.

## 3.2 Verification experiment 2: grounding line evolution

Verification experiment 2 (VE2) is a modified version of VE1, but with part of the region grounded and a net ice flow through
the domain allowed. The setup is identical to VE1 except where stated otherwise in this section. This experiment aims to
combine design simplicity with an evolving grounding line rather than to represent a system directly analogous to a real world
example.

### 3.2.1 Ice component setup

The VE2 initial geometry is given by

$$
\begin{aligned}
z_b &= -20 - 980\left(\frac{x}{100000}\right), & (9)\\
H &= \frac{\rho_{or}}{\rho_i}\left(470 - 400\left(\frac{x}{100000}\right)\right), & (10)
\end{aligned}
$$

where $z_b$ is bedrock elevation relative to sea level and $H$ is ice thickness. Then $z_d$ and $z_s$ are calculated based on floatation and the same densities as in VE1.

The depth dependent inflow ($x = 0$) and outflow ($x = 100$ km for VE2_ER; $x = 99$ km for VE2_EF) boundary conditions for the ice component are given by

$$P_{inflow}(z) \quad = \quad \rho_i g(z_s - z) \tag{11}$$

$$P_{outflow}(z) \quad = \quad \rho_o g z \tag{12}$$

where $P_{inflow}$ and $P_{outflow}$ are pressures prescribed at the inflow and outflow boundaries respectively and $z$ is height relative to sea level (positive up). Zero normal velocity and free slip tangential velocity conditions are imposed at the side walls given by $y = 30$ km and either $y = 0$ (for VE2_ER) $y = -1$ km (for VE2_EF).

The grounding line is allowed to evolve solving a contact problem (Gagliardini et al., 2013). The pressure acting on the underside of ungrounded ice is given by Equation 3.

A sliding relation with a simple effective pressure dependency is used under the grounded ice (Budd et al., 1979, 1984; Gladstone et al., 2017),

$$\tau_b = -C u_b^m z_*, \tag{13}$$

where $\tau_b$ is basal shear stress, $u_b$ is basal ice velocity, $z_*$ is the height above buoyancy (related to effective pressure at the bed, $N$, by $N = \rho_i g z_*$), $m$ is a constant exponent (set to $m = \frac{1}{3}$), and $C$ is a constant sliding coefficient (set to C $= 10^{-4}$ MPa m$^{-\frac{4}{3}}$ a$^{\frac{1}{3}}$).

Height above buoyancy is calculated by:

$$z_* = \begin{cases} H, & \text{if } z_d >= 0 \\ H - z_d \frac{\rho_{or}}{\rho_i}. & \text{if } z_d < 0 \end{cases} \tag{14}$$

This is equivalent to assuming a sub-glacial hydrology system fully connected to, and in pressure balance with, the ocean.

### 3.2.2 Ocean component setup

Ocean bathymetry matches the bedrock prescribed in the ice component (Equation 9). The wet/dry scheme (Section 2.8) is used in this experiment, with a critical water column thickness of $D_{crit} = 5$ m. Ocean potential temperature is initialised at -1.9 °C and salinity at 34.6 on the practical salinity scale.

**ROMS specific details.** The ROMS setup is identical to Verification Experiment 1 except that the baroclinic (slow) time step size is 100 seconds, with 30 barotropic (fast) time steps for every slow time step.

**FVCOM specific details.** The FVCOM model setup is identical to that of Verification Experiment 1.

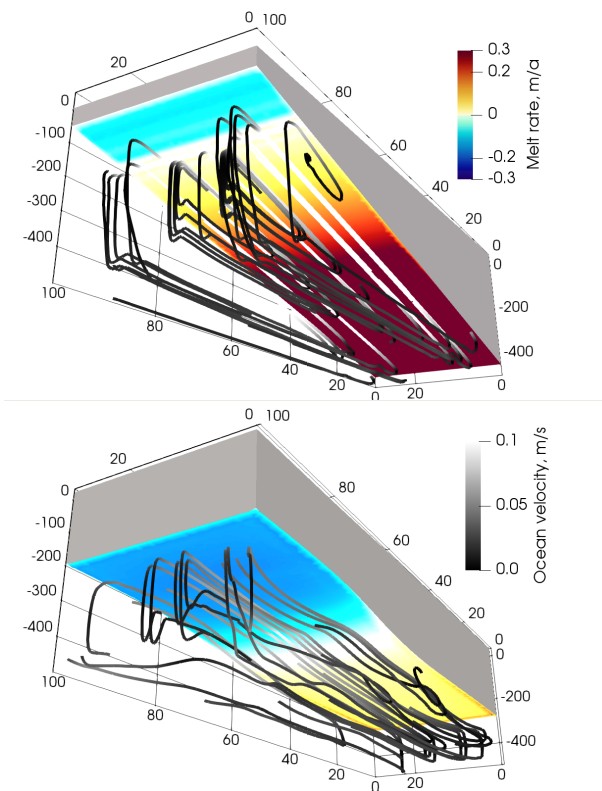

**Figure 4.** Coupled system state after the first (top) and last (bottom) coupling intervals from the experiment VE1_ER (Table 2). The ice shelf is shown in grey, with basal melt rate computed by the ocean shown in colour on the underside of the ice shelf. Ocean streamlines are shown beneath the ice shelf with the grayscale indicating magnitude of simulated ocean velocity. The vertical coordinate is given in m; the horizontal coords are given in km. This was a 100 year simulation.

### 3.2.3 Coupling

The cavity update method for VE2_EF is **Rate** (Section 2.5). For VE2_ER it is **Corrected Rate** with a correction factor of $f_{cav} = 0.01$. With the 10 day coupling interval, this equates to a full correction timescale of approximately 3 years. Other coupling details are as in VE1.

## 4 Verification experiment results

### 4.1 VE1: Floating adjustment

Figure 4 summarises the coupled system state at the start and end of simulation VE1_ER (see also Table 2 for a summary of the experiments). After the first coupling interval (10 days), the ocean component demonstrates a vigorous overturning circulation

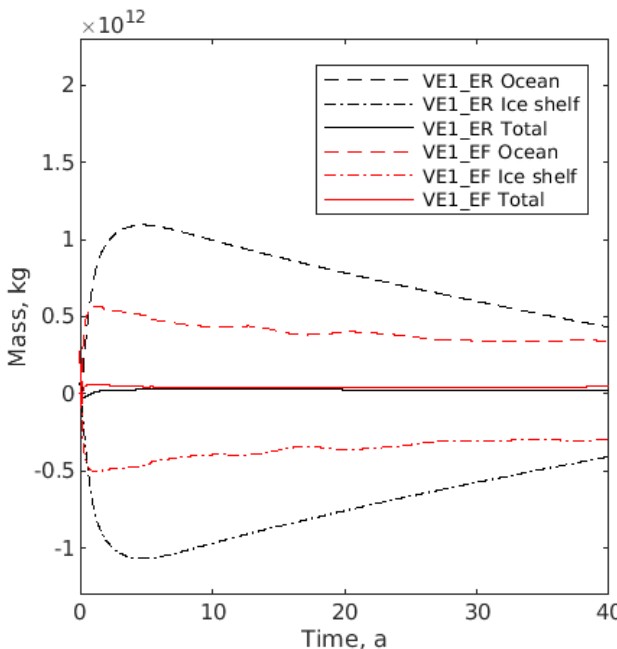

**Figure 5.** Simulated mass evolution over time for the ocean component (dashed lines), the ice component (dash/dot lines) and the total across both components (solid lines) from experiments VE1_ER (black) and VE1_EF (red).

and high melt rates, especially in the deeper part of the domain. After the last coupling interval (100 years) the combination of melting and ice flow has caused a redistribution of the ice shelf, with an overall reduction in the along-domain gradients. The melt rates and overturning circulation are much weaker than at the start.

The ocean circulation throughout the simulation is predominantly a buoyancy driven overturning along the domain, with very little cross-domain flow. The peak ocean flow speeds are always located at the top of the ocean domain directly under the 455 ice shelf, where a fast, shallow buoyancy driven flow from deeper to shallower ice draft is balanced by a much deeper return flow.

Figure 5 shows the evolution over time of the total mass of both ice and ocean components and the total coupled system from experiments VE1_ER and VE1_EF. Note that both ocean models employ the Boussinesq approximation, and that the mass in Figure 5 is calculated as volume multiplied by the reference ocean density from Table 2. Relatively rapid mass transfer from 460 the ice to the ocean occurs during the first few years as the relatively warm ocean water transfers its energy to the ice. After this initial period of net melting, the ocean water temperature is close to freezing point, and a long term freezing trend can be seen, stronger and more sustained in the ROMS ocean component than FVCOM. In a physically realistic coupled system, the ice and ocean would come into thermodynamic equilibrium and the spatial net mass transfer would approach zero.

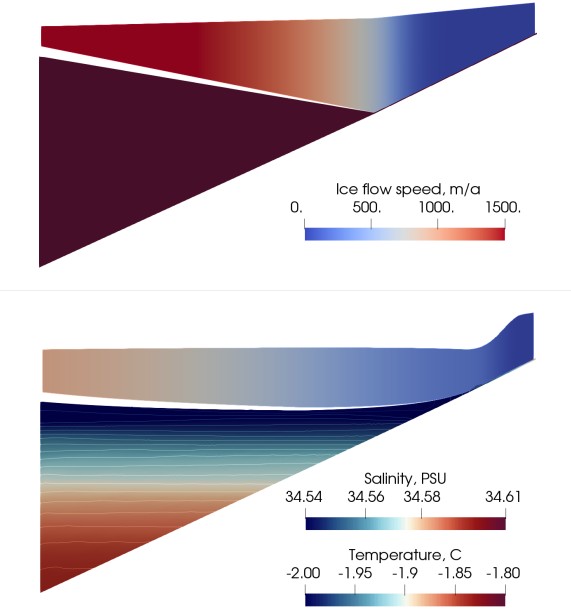

**Figure 6.** Profiles through the center line for experiment VE2 after the first ice component time step (top) and after 25 years (bottom). Ice flow speed is shown (flow direction is right to left). Ocean temperature (solid colour) and salinity (contours) are shown after 25 years (these are uniform at the start of the run, hence the solid colour for the ocean in the upper plot). Vertical exaggeration is 50 times. The gap between ocean and ice shelf is half an ocean grid cell and is a plotting artefact (The upper extent of the plotted region for the ocean is the uppermost rho point, which is half an ocean grid cell below the top of the ocean domain).

The net mass change of the coupled system is more than an order of magnitude smaller than the mass change of the individual components for both experiments VE1_ER and VE1_EF. The current study does not use conservative regridding (Section 2.3), and so machine precision conservation is not expected. There are additional potential sources of error. The lag of ocean component behind ice component (Section 2.1.1) will cause a similar lag in total mass evolution. Use of the "**Corrected rate**" cavity option (Section 2.5) prioritises geometry consistency between components above mass conservation. The aim of analysing mass conservation in the current study is to ensure that the cumulative impact of these potential error sources is small compared to the signal. This has been achieved, and it will be possible to quantify and minimise or eliminate all sources of error in future studies using conservative regridding methods.

## 4.2  VE2: grounding line evolution

This is a partially grounded experiment in which the ice component boundaries are not closed, a through-domain flow of ice is allowed, and the grounding line is allowed to evolve in the coupled system (described in Section 3.2). While the initial slope of the lower surface of the ice shelf is the same in both VE1 and VE2, the open inflow and outflow boundaries in the ice

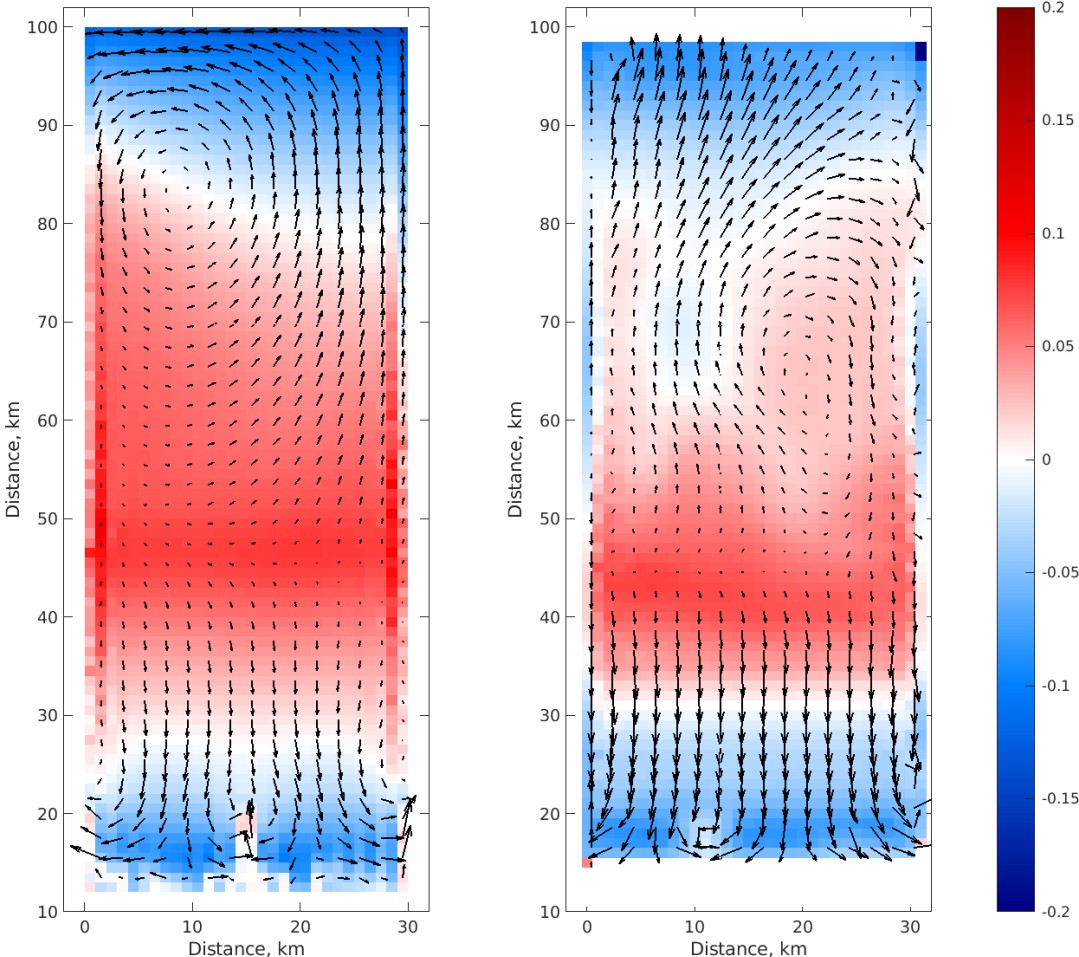

**Figure 7.** Ocean horizontal velocities in the upper layer (black arrows) and basal melt rate (red indicates melting, blue refreezing) after 25 years of simulation VE2_ER (left) and VE2_EF (right). Outputs on the FVCOM mesh were regridded onto a 1km regular grid. Both FVCOM and ROMS outputs were subsampled at 2km resolution for this plot.

component and the relatively shallower ice in the grounded region both lead to a shelf that is much shallower in slope for VE2 than for VE1 for most of the simulation period. Figure 6 illustrates the shape of the ice sheet/shelf at the start of the simulation and after 25 years (from simulation VE2_ER). Note that the ice outflow boundary is more active than the inflow, with the flux into the domain through the inflow boundary remaining small and positive throughout the simulation. The ice draft is deepest 480 in the middle of the domain, at around 30km downstream (in terms of ice flow direction) from the grounding line. The ice draft

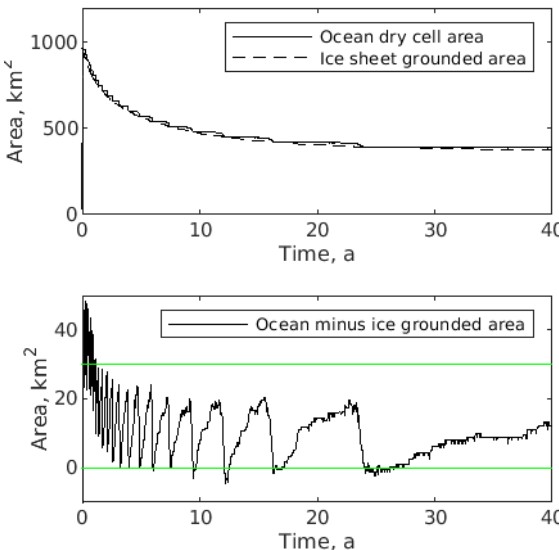

**Figure 8.** Top: A comparison of grounded area in the ocean component (total area of dry cells) against grounded area in the ice component (total area of grounded elements). Bottom: The difference between ocean and ice grounded area. These are from simulation VE2_ER. The green lines are drawn such that their distance apart is equivalent to the area of one row of ocean grid cells.

impacts on circulation and melt, with the strong overturning of VE1_ER not present here. Melting occurs under the deepest ice, with refreezing elsewhere (Figure 7).

Comparing the coupled simulation VE2_ER to the ice-only simulation (not shown) where the only difference is that the ice component features zero basal melt, it might be expected that the coupled simulation would exhibit a significantly thinner ice

shelf due to melting. However, the ice dynamics partially compensate for this in terms of the ice geometry: the melt-induced thinning leads to acceleration in the ice and the thickness difference is smaller than expected. However, this should not be interpreted as a stabilising feedback response of ice dynamics to ocean induced melting, as the increased ice flow would tend to drain the grounded ice more quickly, potentially triggering marine ice sheet instability (Schoof, 2007). Instead this effect may tend to partially mask an ocean-induced ice sheet destabilisation if the observational focus is on ice shelf geometry.

As described in Section 2.8, the ice and ocean component each evolve the grounding line on their own time step and on their own grid or mesh. There is potential for discrepancy between ice and ocean grounded area due to method of cavity evolution (Section 2.5), regridding errors, the inherent differences between grids or meshes, and the methods used to determine grounding line position. While ice geometry is a key determinant of grounding line position, the ice component also tests for a contact force (Gagliardini et al., 2013) and the ocean component ROMS tests height above buoyancy against the free surface variable

$\zeta$ (Section 2.8). Here we look at consistency of grounded area between components.

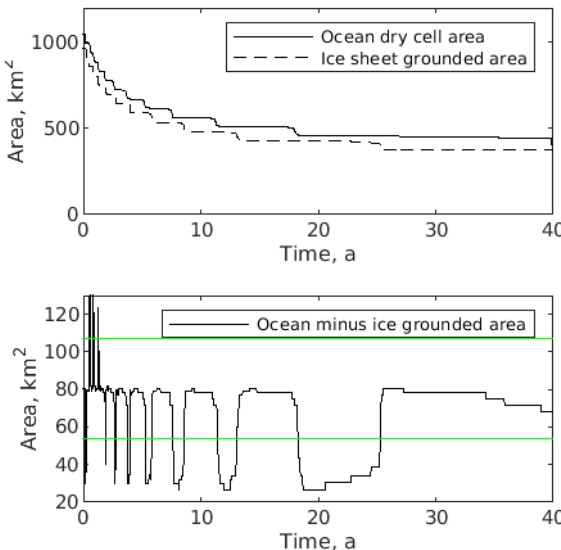

**Figure 9.** Top: A comparison of grounded area in the ocean component (total area of dry elements) against grounded area in the ice component (total area of grounded elements). Bottom: The difference between ocean and ice grounded area. These are from simulation VE2_EF. The green lines are drawn such that their distance apart is equivalent to the area of one row of ocean elements.

The evolution of grounded area in both ice and ocean components is shown in Figure 8 for simulation VE2_ER. While the ice component employs an unstructured mesh of triangular elements (on the lower surface of the 3D ice body), the ocean component employs a regular grid of square cells. The ocean component appears to exhibit a step-like reduction over time of grounded area. This is due to the row-by-row manifestation of grounding line retreat in the ocean component due to the

alignment of grid rows with the linear downsloping geometry. Grounding line retreat starts at the lateral edges of a row (ungrounding near the sidewall boundary) and the "wetting" of dry cells propagates toward the centre of the row. This step-like behaviour (with the spacing of the green lines in Figure 8 indicating the total area of a row of cells) explains the main difference between ice and ocean grounded area. The evolution of grounded area is shown in Figure 9 for simulation VE2_EF. Behaviour is similar to VE2_ER.

The initial rapid reduction in grounded area is due to the initial geometry. A region immediately upstream of the grounding line is initially very lightly grounded, and this region quickly becomes floating. The ocean component lags the ice component in this un-grounding, as can be seen in the first part of the difference plot in Figures 8 and 9. This lag is in part due to the "**Rate**" and "**Corrected rate**" cavity update methods, in which the ocean component uses the most recent two ice component outputs to calculate a rate of change of geometry. This inevitably causes the ocean component to lag by approximately one

coupling interval. The discrepancy may also be in part due to the fact that the region in question is close to floatation, thus the

threshold for dry cells to become wet is highly sensitive to $\zeta$, at least for the ROMS implementation. In both experiments, the ice - ocean grounded area discrepancy has a tendency to reduce over time.

The computational time spent in both the ice and ocean components was measured for simulation VE2_ER. The ice component is more expensive than the ocean component during the first coupling interval, but is significantly cheaper thereafter. Total time spent in the ice component over the 46 year simulation is approximately one third that spent in the ocean component. The computational time spent within the central coupling code (calling routines and regridding) was negligible compared to time spent in ice and ocean components. This is with a 10 day coupling interval. If fully synchronous coupling is approached (i.e. if the coupling interval approaches the ocean time step size) the ice component will become much more expensive and it is possible the central coupling code may become significant. We do not anticipate fully synchronous ice-ocean coupling to become practical in the near future, at least not if the ice component directly solves the Stokes equations without simplifying assumptions, as is the case in the current study. The fully synchronous coupling of Goldberg et al. (2018); Snow et al. (2017) is achieved by using the "shallow shelf approximation" for the ice component and running both components on the same grid.

## 5   Conclusions

We have presented a flexible coupling framework for ice sheet/shelf and ocean models which allows the user to choose between different ice and ocean components. We have demonstrated the functioning of this framework in simple test cases, both with and without a moving grounding line. We have demonstrated conservation of mass and consistency of grounding line evolution using semi-synchronous coupling.

FISOC provides run time variable exchange on the underside of ice shelves. Providing such variable exchange at vertical ice cliffs, which are more common in Greenland than in Antarctica, will require minor developments to the coupling code, but the ocean components currently coupled through FISOC may need more significant developments in order to represent the buoyant plumes rising up ice cliffs.

Our coupled modelling framework is suitable for studying Antarctic ice sheet/shelf – ocean interactions at scales ranging from investigations of ice shelf channels (features with a spatial scale of typically a few km) up to whole Southern Ocean - Antarctic Ice Sheet coupled evolution. We are currently setting up simulations across this range of scales to address key processes surrounding Antarctic Ice Sheet stability and sea level contribution.

*Code availability.* The coupled modelling code is available from the FISOC github website https://github.com/RupertGladstone/FISOC under the GPL2 licence. The exact version of the FISOC code used to produce the results used in this paper, along with exact versions of ice and ocean model components, is given by code urls and specific commit hashes in a README file. This README file is archived along with the required input files on Finnish HPC machine Allas, and is publicly available from the Allas website https://a3s.fi/COLD_share/FISOC_GMD_files.tar.gz.

*Author contributions.* RG led development, implementation of experiments, and paper writing. BGF, DG, QZ, TH, DS, SM, CZ, LJ, XG, KP, TZ contributed to development and or testing. BGF, DG, QZ, TH, CZ, YX, TZ contributed to implementation of experiments. All authors contributed to paper writing.

*Competing interests.* No competing interests are present.

*Acknowledgements.* Rupert Gladstone was funded from the European Union Seventh Framework Programme (FP7/2007-2013) under grant agreement number 299035. This research was supported by Academy of Finland grants 286587 and 322430. The authors wish to acknowledge CSC - IT Centre for Science, Finland for computational resources. Tore Hattermann acknowledges financial support from Norwegian Research Council project 280727. Qin Zhou acknowledges financial support from Norwegian Research Council project 267660. Konstantinos Petrakopoulos's work was supported by CSLC grant G1204 from NYU Abu Dhabi.

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
