# Peer review of "A Framework for Ice Sheet - Ocean Coupling (FISOC) V1.1"

_Geoscientific Model Development, 2020_

## Referee Comment (RC1) · Anonymous Referee #1 · 6 Sep 2020

Gladstone et al. present a new coupling infrastructure (FISOC) that is designed to accommodate various types of ocean and ice-sheet models. The manuscript is well written, and FISOC, the verification protocol as well as the results are clearly presented. By implementing two ocean models (with distinct regridding methods) within the FISOC framework, the authors demonstrate the flexibility of their approach. The two simple verification experiments proposed here are convincing and could be used for the verification of other coupled models. Given the several references to synchronous vs asynchronous approaches in the paper, I would have found it useful to see a set of sensitivity experiments to estimate the influence of the coupling interval (e.g., from 1 day to a few months), which would also further demonstrate the flexibility of FISOC, but this is probably beyond the scope of the present paper. Therefore, besides a few

elements that require clarifications (see below), I recommend the manuscript for publication in GMD.

————-

Minor comments:

- L.7: "thesemechanisms"

- L.29: other anterior references for the ice geometry feedback onto melt rates include De Rydt et al. (2014), Timmermann and Goeller (2017) and Donat-Magnin et al. (2017).

- L. 43-45: "offline coupling" and "partial restart" should be defined. I am not sure that "offline coupling" is very relevant as deciding what is "online" and "offline" can be somewhat subjective and the synchronous/asynchronous distinction is probably enough.

- L.102: there is a subsection 2.1.1 but no subsection 2.1.2. I suggest putting everything under 2.1 with no 2.1.1 subsection.

- L.113: indicate that the user manual is provided on github or as supplementary file.

- L.135-138: in which circumstances is it necessary to extrapolate? Is the ocean grid extends beyond the ice grid, I would expect the ice geometry (seen by the ocean) to be taken from observational data, not extrapolated from the ice model.

- L.151: ". Cavity geometry..." add "seen by the ocean".

- L.161: It is unclear what is meant by "partial restart" here. In Favier et al. (2019), the ocean is restarted every coupling interval by conserving its velocity field (and all information on the current and previous time steps that is usually used to restart the ocean model), not only temperature and salinity as in De Rydt et al. (2016), so shouldn't it be called a full ocean restart. The main difference between Favier et al. (2019) and ROMS/FVCOM here is that the ice geometry seen by the ocean evolves by step, i.e. every coupling interval rather than every ocean time step, and that there is an associated correction on barotropic velocities to cope with that.

- L.311-312 (or in section 2.6): please specify whether the 3-equation formulation is used with a constant exchange velocity. It could be worth mentioning that "rotation is disabled" to avoid asymmetric melt rates and seek a perfectly flat solution for the ice draft.

- L.320-323: if these details are given for FVCOM, I guess they should be provided for ROMS as well. Having said that, I am not sure these are useful in this paper, unless there are reasons to think that some schemes are less conservative than others.

- L. 311 and 353-354: are these in-situ or potential temperatures?

- section 4.1: it would be worth mentioning whether the ocean models make the Boussinesq approximation (which would probably mean that they are expected to conserve volume rather than mass). Is the ocean mass in Fig. 4 derived from the uniform sea water densities shown in Tab. 2 based on its volume?

———-

Additional references

De Rydt, J., Holland, P. R., Dutrieux, P. and Jenkins, A. (2014). Geometric and oceanographic controls on melting beneath Pine Island Glacier. Journal of Geophysical Research: Oceans, 119(4), 2420-2438.

Donat–Magnin, M., Jourdain, N. C., Spence, P., Le Sommer, J., Gallée, H. and Durand, G. (2017). Ice–Shelf Melt Response to Changing Winds and Glacier Dynamics in the Amundsen Sea Sector, Antarctica. Journal of Geophysical Research: Oceans, 122(12), 10206-10224.

Timmermann, R. and Goeller, S. (2017). Response to Filchner–Ronne Ice Shelf cavity warming in a coupled ocean–ice sheet model–Part 1: The ocean perspective. Ocean Science, 13, 765-776.

---

## Referee Comment (RC2) · Xylar Asay-Davis (Referee) · 13 Oct 2020

**Review of Gladstone et al. "A Framework for Ice Sheet - Ocean Coupling (FISOC) V1.1"**

Reviewer: Xylar Asay-Davis

I wish my name to be relayed to the authors, as I do not support the practice of anonymous review.

**General Comments:**

This paper describes a framework, FISOC, built on the Earth System Modeling Framework (ESMF), for coupling ice sheet and ocean components. Since ESMF is used for coupling in many Earth System Models (ESMs), the authors suggest that FISOC could provide an important stepping stone toward ice sheet-ocean coupling in an ESM. The paper describes the coupling infrastructure as well as the ice sheet and ocean components used for coupling verification. Then, the authors use two idealized test cases to demonstrate approximate conservation of mass and consistency between the grounding line as represented in each component.

The paper is well written and well organized. The figures do a good job at illustrating the design concept of the framework and its flexibility in addressing the unique requirements of the components it currently supports. For the most part, I find the description of FISOC and the verification experiments appropriately detailed and easy to follow. There are a few areas where I think more clarification or detail would be useful, as detailed below in the specific comments, before the paper is ready to publish in GMD.

First, the text mentions briefly that FISOC currently uses "sequential coupling" (but that "concurrent parallelization" would presumably require minimal effort). However, the text does not provide sufficient detail on how sequential coupling is performed, in particular what the conceptual start and end time of each component's coupling interval is. Nor is there any description of how this might be different for concurrent parallelization. I think these are needed to better understand the coupling strategy and, in particular, the inconsistencies between the geometry as represented in each component.

Second, there is no discussion in the paper about how the ocean components ensure ocean connectivity (if at all) and how this may need to be accounted for in the coupling. Would the ocean components allow melting in "subglacial lakes" (as emerge in the experiments of De Rydt and Gudmundsson, 2016) that meet the flotation criterion but are not connected to the ocean? If, so, this could drive unrealistic ice-sheet dynamics. Would these subglacial lakes be considered part of the floating area in the ocean component? If not, would this lead to a significant discrepancy in the geometric representation (or at least accounting) between the two components?

Third, while emphasis is placed on conservation, the interpolation methods used in the VE1 experiment are not conservative and therefore would not be appropriate for flux fields (like

the melt rate) in ESMs. Relatedly, the results from VE1 demonstrate approximate, but not machine-precision, conservation of mass. Could this be improved by using conservative interpolation and also accounting for the mass accumulated in the coupler during a coupling interval?

Fourth, I found the geometry and design choices of the VE2 experiment hard to follow. It would be helpful to have a figure showing the initial side-view (x-z) geometry for the experiment as well as a cross-section of the geometry at 25 years shown in x-y in Fig. 5. It would also be helpful to have a velocity plot for the ice-sheet component similar to that for the ocean components in Fig. 6. The behavior of VE2 strikes me as quite dissimilar to realistic ice sheet/ice shelf dynamics in that the thickest part of the ice shelf is in an ungrounded region and (as near as I can tell) ice seems to be flowing out of the "inflow" boundary at x = 0. It seems like some acknowledgement of the rather significant limitations of this experiment are needed somewhere in the discussion.

As long as the authors make an effort to address these comments or explain their reasoning for not addressing them, I do not need to see a revised manuscript before publication.

**Specific Comments:**

l. 53: ESMF does not need to be redefined, since it was already defined on l. 47-48

l. 69-70: I think this is an important point that should be explored in a new subsection of section 2. Presumably, both components start at t=0. Which component runs first? Let's say it's the ocean. Once the ocean has finished a coupling interval, it has computed a melt rate. Is this averaged over the coupling interval or is the instantaneous value at the end of the coupling interval used? (This has important implications for how precise conservation of mass will be computed.) Presumably, the dD/dt is initialized to zero in the ocean component, so this is clear for the first coupling interval, but I will come back to this.

Then, let's say the ice-sheet component runs. It is able to apply the known melt rate for the coupling interval (10 days in VE1 and VE2), so there is no conceptual time lag here if the melt rate is a time average but there is one if it is an instantaneous value from t = 10 days. Based on the results of this coupling step, dD/dt can now be computed and interpolated to the ocean grid or mesh. This dD/dt is time-centered at t = 5 days, but will be applied over days 10 to 20 in the ocean component, so this is the source of a time lag that you discuss later.

If the components run in the opposite order, it is conceivable that the time lag could be placed on the melt rate instead of dD/dt. There is no explicit description of this, but I get the sense that this was not the choice that was made, since results from VE2 discuss a lag in D, not in melt rate.

If I am correct in assuming that the ice sheet component updates second, after the ocean, in a given coupling interval in the sequential scheme, this likely means you do not need to

account for mass that conceptually accumulates in the coupler during a coupling interval. It seems important to me to discuss that "concurrent parallelization" will require a time lag in both dD/dt and in melt rate, since each component will be updated based on the state (or time average) at the end of the previous coupling interval. In this scenario, it would be important to keep track of the mass that conceptually accumulates in the coupler over a coupling interval over a time step. This is the approach used for fluxes between components in the Community Earth System Model (CESM) and Energy Exascal Earth System Model (E3SM), the ESMs I am most familiar with, and I think in other ESMs as well. Even in cases where components may run sequentially on the same processors, I do not think it is common to take advantage of this to remove the time lag in fluxes between components because of the conservation issues that could arise.

Again, I feel like some discussion of these nuances is missing from Sec. 2.

l. 85-86: ESMF does not need to be reintroduced and the citations are not needed because the acronym is already defined on l. 47-48 and the citation are already covered on l. 69-70.

l. 134: "All FISOC simulations to date have used a Cartesian coordinate system." Is Elmer/Ice capable of using a spherical coordinate system? The BISICLES and MALI models that I have worked with both work only on Cartesian (polar stereographic) meshes, which requires special care to ensure flux conservation but can be handled as long as the coupling infrastructure is aware of the discrepancy in areas between the component models.

l. 155: "FISOC assumes that time-step sizes are not adaptive." This seems quite restrictive to me and potentially unnecessary. It seems like this could use some discussion. I have worked with the BISICLES and Parallel Ocean Program coupled model called POPSICILES. In that model, we had a different coupling strategy and we always ran with concurrent parallelism over a coupling interval. We found many cases doing more realistic simulations where it was highly beneficial that BISICLES (which can perform adaptive mesh refinement) could refine its time step to handle a particularly tricky geometric configuration that might emerge spontaneously. We simply required that BISICLES perform a time step that exactly reached the coupling end time as the last step in a coupling interval. It seems like this strategy would also be compatible with FISOC, and therefore the requirement that the coupling interval is equal to the ice-sheet time step is unnecessarily restrictive. If there are important reasons for the restriction, it would be helpful if they are clarified. If this is not a strict requirement but rather has been the convention in simulations to date, this should be discussed.

Eq. (1): I think some more nuanced discussion is needed about the time-centering of dD/dt (which is at t - ½ Delta t) and when the dD/dt is actually applied in the ocean component (centered at t + ½ Delta t). You have dD/dt subscripted at time t but I do not think that is correct in either component.

l. 180: It is important to clarify that D is positive up. This might seem obvious from an ice-sheet modeling perspective, where D being positive down would not be an obvious choice since it can take either positive or negative values. But D in the ocean is often used

for "depth" and is almost universally a positive quantity, so the choice of variable names and the sign convention are not intuitive for ocean modelers. There are also some later equations where I think the sign of D is not correct (as I will point out), leading to further confusion about the sign convention. A lot of confusion in the paper might be spared by renaming this variable "z_d" to go with "z_b" for the bedrock elevation/bathymetry.

l. 183-184: "but has the potential for the ice and ocean representations to diverge over time as a result of regridding artefacts": Isn't some part of the divergence in time likely to come from the fact that there is a time lag between dD/dt from the ice component and as applied in the ocean component?

l. 213: "...FISOC can pass the temperature gradient from the ice component directly to the ocean component." I think this requires some discussion. The temperature at the ice-ocean interface is computed on the ocean time step and could potentially have significant temporal variability within a coupling interval (e.g. because of ocean eddies). Would the ice sheet get the time-average of this field as one of the coupling fields, and use this to compute the temperature gradient? If so, this would result in the temperature gradient going back to the ocean having a time lag of 2 coupling intervals in the temperature at the ice-ocean interface. Maybe this doesn't matter.

An alternative approach would be to pass the temperature in the bottom ice layer (and the ice thickness) and allow the ocean model to compute the temperature gradient on its time step. The differences between these approaches is likely only to matter for particularly long coupling intervals but it might still be worthy of some though and some discussion. The choice is not entirely obvious, at least not to me.

Eq. (3): I believe the RHS of this equation needs a negative sign if D is positive up. Otherwise, the pressure would be negative.

Eq. (4): There is a sign problem with this equation, too. I am pretty sure it is that the whole RHS needs a negative sign again. If drho_o/dz were 0, you expect a positive pressure for a negative D. Later, drho_o/dz is given as a positive number, which is not physically reasonable if z is positive up. Density should decrease toward the ocean surface. But if that term is positive, pressure should increase because of increasing density at depths, so the term -0.5 drho_o/dz D_[O] is positive for negative D_[O], as it should be. If drho_o/dz is changed to be negative (as I think it should be), the sign of this term would also need to be flipped. In any case, there's something to be fixed here. The confusion may arise from an ocean component that uses a positive-down definitely of z, but I think the paper needs to pick positive-up for everything and stick with it.

l. 245: "z_b is the bottom boundary depth (bathymetry, aka bedrock depth)": Most times the term "depth" is used in ocean modeling, there is an implication that it is positive-down. The fact that the variable is called "z_b" might tend to counteract that but I would state explicitly that it is positive-up.

l. 246: "D_crit is a critical water column thickness (or depth)": This one is strictly positive, and is unrelated to D, which I find pretty confusing. Again, renaming D to z_d would do a lot to help with this. By the way, I don't see how the "or depth" bit applies at all in this context.

Eqs. (6) and (7): I'm having trouble following these. An illustration would help a lot, but some text carefully defining the variables involved might do the trick.

The original definition was "η is the free surface variable". In the new context, it's pretty hard to understand what η is. The best way I can understand it is that η + D_[O] is the ocean's representation of the location of the ice-ocean interface, which is allowed to move up and down because of changes in ocean dynamic pressure. Maybe some explanation along these lines would be helpful.

As far as I can tell, Eq. (7) is equivalent to Eq. (6) except that D_crit is now a minimum ice-sheet thickness below flotation rather than a minimum ocean-column thickness? This is confusing and needs some explanation as to what exactly it means and why ROMS chose this definition instead of the simpler one from FVCOM. It is confusing to use the same name for variables with distinct meanings in the two models. Also, shouldn't a slightly different D_crit be used for ROMS to get the same ocean-column thickness (assuming this is desired)?

l. 257: "as described in Section 2.8": The current section is 2.8, so this must be a mistake. Maybe the reference is supposed to be to Sec. 2.5?

l. 272-274: "The coupling is purely geometric in that the ocean component passes an ice shelf basal melt rate to the ice component and the ice component passes a rate of change of ice draft to the ocean component." This may be a nuance of interpretation but I do not think of the mass flux in the form of a melt rate as being a geometric quantity, so I would disagree that the coupling is purely geometric.

l. 280-282: I think it would be helpful to have an explanation for why the FVCOM simulation required a domain of a slightly different size. It is not clear if the sizes given in Table 2 are for both components or just the ocean component (in which case a row is needed for the ice component).

l. 295: "$\rho_{or}$ = 1027 kg m^−3": You give a slightly different value for FVCOM in Table 2 but this difference is not addressed here or anywhere else. Why the difference?

l. 297-299 and Eq (9): You gave a definition of the pressure at the interface in Sec. 2.7 already, and it was different from this for FVCOM. Presumably this redundant definition is not needed.

l. 303: I think "zero net accumulation" is a slightly confusing phrase here. I assume the idea is that ice sheet models typically have a field of net accumulation (called a) and that this is zero everywhere in this case. But it lends itself to the misunderstanding that there is

accumulation but that the net effect is zero (e.g. averaged in time, space or both). Could this be simplified but just removing the word "net"?

l. 315 "ROMS specific details." Nothing is given about vertical mixing or eddy parameterizations, whereas these details are given for FVCOM.

No details are given about how the three-equation parameterization is handled in either ocean component. For example, where are "far-field" temperature and salinity sampled? What parameters are used? (Are they the same for both models?) Which equation of state and equation for the freezing point is used in each.

l. 317: "FVCOM specific details." In addition to the above, no details are given about time stepping for FVCOM as they are for ROMS.

Eqs. (10) and (11): As I stated in my general comments, I think a figure is needed to help better understand this experiment. A starting point would be a side-view (x-z) figure showing the initial ice-sheet, ice-shelf and ocean configuration as given by these equations.

Also, since rho_or is slightly different for the 2 ocean components, is (11) accurate and H is therefore slightly different for the two but D is the same?

l. 338: "No restrictions to ice flow are imposed at the upstream and down stream boundaries". I have several difficulties here. First, some more explanation is needed about what "no restrictions" really means. Presumably, this means that ice is free to flow out of the boundaries. I do not see how ice can flow in through these boundaries if there is "no restriction". Is it necessary to calculate stresses at the boundaries and, if so, how is this handled (in particular driving stress)? Why was an open boundary condition like this chosen at x=0? A more typical setup would place a solid boundary here so it acts as something of an ice divide. This would also make the direction of ice flow a lot less ambiguous.

That brings me to the second point, which I will discuss more below. The ice flow field is not discussed but I get the impression based on the thickness evolution that flow is happening out of both the x=0 and x=100 km (or 99 km) boundaries, so that the "upstream" and "downstream" directions aren't well defined in this problem.

l. 364-366: I found this paragraph redundant to the paragraph on l. 268-270 and subsequent text. I realize it is nice to summarize things again from previous sections but this seems too repetitive to me.

l. 372-373: I don't think "along the domain" and "cross-domain" are well defined directions because they take the perspective of the ice flow in a context where ocean circulation is being discussed. I would just call these the x and y directions.

l. 382-383: "The net mass change of the coupled system is more than an order of magnitude smaller than the mass change of the individual components for both experiments VE1_ER and VE1_EF." I think this needs considerably more discussion. For ESMs, anything less than machine-precision conservation is not considered acceptable and is one of the most important mechanisms for diagnosing model inconsistencies. To accomplish this, conservative regridding is always used for flux fields (the melt rate in the case of FISOC, and the heat flux in the future).

Ice sheet-ocean modeling requires that special care must be taken to distribute that flux field to the ice-sheet component because melt should not get distributed to grounded cells by mistake but it also should not be lost in the regridding process because this would affect conservation. This issue is exacerbated by inconsistent representation of the grounding line between components. Is this taken into account in FISOC? If so, please discuss. If not, please discuss this as a potential issue for future consideration.

Aside from interpolation, conservation of mass may be inexact in FISOC because of the lag between when melt rates are computed in the ocean component and when they are applied in the ice component. I convinced myself when I was discussing the staggered coupling approach above that this is likely not the case in the current approach but it would be in an approach with concurrent parallelism. Even so, it would be important to diagnose that total mass going into the coupler is exactly equal to total mass coming out of the coupler after each coupling interval (i.e. after both components have run) or that the difference between these two is computed and stored within the coupler to be distributed appropriately at the next coupling interval.

Overall, I would like to see some discussion about why conservation of mass in FISOC is good but not machine-precision good.

l. 386-387: "While the initial slope of the lower surface of the ice shelf is the same in both VE1 and VE2": I misunderstood this the first time I read it to be saying that the D's for VE1 and VE2 were the same. They differ by 20 m but the slope is the same. I guess it's fine as it is but I wanted to let you know about the confusion, in case you want to do anything about it.

l. 387: "the open inflow and outflow boundaries": I remain confused about the open "inflow" boundary at x = 0. Is it really inflow? If so, how does open inflow work?

Fig. 5: First, as I stated in my general comments and as I think you are fully aware, this is an odd ice-shelf geometry. It is also not very intuitive to see thickness plotted as an x-y field, at least not for me. It would be more helpful in my opinion to have a more 3D plot similar to Fig. 3. It would also be really helpful to have a vector field for the ice like the one for the ocean velocity in Fig 6, especially because I want to see how much the weird geometry is due to outflow (instead of inflow) at x = 0.

Fig. 6: Are these fields interpolated to a common, regular grid? They look like they might be and, if so, this should be mentioned.

l. 415-421: It may be worth remarking that the difference in grounded area does not increase with time even with the Rate method, at least in this case.

**Typographical and grammatical corrections:**

l. 7: "these mechanisms" missing a space

l. 8: a comma is needed between "this" and "ocean"

l. 22: "(MISI) (Mercer, 1978; Schoof, 2007; Robel et al., 2019)": I would combine these parentheses as you have done elsewhere: "(MISI; Mercer, 1978; Schoof, 2007; Robel et al., 2019)"

l. 38: Similar to above: "(ISOMIP+; Asay-Davis et al., 2016)"

l. 46: commas are needed: "Here, we present a new, flexible…"

l. 46: for consistency, a semicolon is needed instead of a comma: "(FISOC; Section 2)"

l. 47-48: "Earth System Modeling Framework" is a proper name so I think it needs to be spelled with the American version of "Modeling" that is used on their website: https://www.earthsystemcog.org/projects/esmf/

l. 53-54: "(Hill et al. (2004); Collins et al. (2005))" should be "(Hill et al. 2004; Collins et al. 2005)" without the nested parentheses.

l. 105: a comma is needed between "versa)" and "all"

l. 117: a comma is needed between "cases" and "a non-standard"

l. 120: "Regional Ocean Modeling System" is also spelled with the American spelling of "Modeling" in the documentation I could find: https://www.myroms.org/

l. 120: There should not be nested parentheses: "(ROMS; Shchepetkin and McWilliams 2005)"

l. 120-121: For consistency, this should be "terrain-following, sigma-coordinate" (with a comma and a second hyphen). In general hyphenation is used a lot more sparsely in this writing than I would use it but I fully acknowledge that that is a stylistic choice.

l. 123: Remove the nested parentheses: "(FVCOM, Chen et al. 2003)

l. 136: a comma is needed between "extrapolation" and "which"

l. 142: "time step" is not typically hyphenated but this may be a stylistic choice.

l. 174: a comma is needed between "used" and "this"

l. 176: a comma is needed between "large" and "occasional"

l. 194: a comma is needed between "case" and "the user"

l. 209: "ocean model ice shelf cavity shape" is quite a long compound noun…

l. 228: "kg m" needs a space or half-space

l. 247: a comma is needed between "Thus" and "cells"

Eq (7): Please remove the asterisk as a multiplication symbol.  It is not needed and is not considered a valid multiplication symbol (outside of code).

l. 258: a comma is needed between "Study" and "dD/dt"

l. 275: I've left most of the hyphenation choices alone but I feel pretty strongly that "uniform-thickness" should be hyphenated.

l. 286: a comma is needed between "system" and "we"

l. 287: a comma is needed between "Therefore" and "the"

l. 288: a comma is needed between "corners" and "where"

Eq (9): should end in a comma, not a period.

l. 309: a comma is needed after "experiment" at the end of the line

l. 326-327: commas are needed after "VE1_ER" and "VE1_EF"

l. 338: "down stream" should be "downstream"

l. 360: a comma is needed between "interval" and "this"

l. 368-369: a comma is needed between "days)" and "the" and again between "years)" and "the"

l. 378: a comma is needed between "melting" and "the"

l. 380: a comma is needed between "system" and "the"

l. 388: commas should be removed from "...component and the relatively shallower ice in the grounded region both…"

l. 397: a comma is needed between "melting" and "as"

l. 400: a comma is needed between "2.8" and "the"

---

## Author Comment (AC1) · 23 Nov 2020

**Author response, "A Framework for Ice Sheet - Ocean Coupling (FISOC) V1.1"**

**Rupert Gladstone et al**

**November 23, 2020**

We repeat the reviewer's text in black and provide our response in blue font.

**1  Response to reviewer 1, anonymous**

Gladstone et al. present a new coupling infrastructure (FISOC) that is designed to ac- commodate various types of ocean and ice-sheet models. The manuscript is well written, and FISOC, the verification protocol as well as the results are clearly presented. By implementing two ocean models (with distinct regridding methods) within the FISOC framework, the authors demonstrate the flexibility of their approach. The two simple verification experiments proposed here are convincing and could be used for the verification of other coupled models. Given the several references to synchronous vs asynchronous approaches in the paper, I would have found it useful to see a set of sensitivity experiments to estimate the influence of the coupling interval (e.g., from 1 day to a few months), which would also further demonstrate the flexibility of FISOC, but this is probably beyond the scope of the present paper. Therefore, besides a few elements that require clarifications (see below), I recommend the manuscript for publication in GMD.

We thank the reviewer for the positive review and for the comments that will improve clarity of the manuscript.
There are many interesting idealised studies to be carried out with FISOC and other ice sheet - ocean coupling tools, and we agree that impact of the coupling interval is one of these. In fact we are currently conducting such experiments and hope to be able to report on them in a future publication. We also agree that this is beyond the scope of the current paper.

Minor comments:
- L.7: "thesemechanisms"
Corrected, thanks.
- L.29: other anterior references for the ice geometry feedback onto melt rates in- clude De Rydt et al. (2014), Timmermann and Goeller (2017) and Donat-Magnin et al. (2017).

The Timmermann and De Rydt references are the most relevant mentioned by the reviewer, and we have incorporatd these into the introduction.

- L. 43-45: "offline coupling" and "partial restart" should be defined. I am not sure that "offline coupling" is very relevant as deciding what is "online" and "offline" can be some- what subjective and the synchronous/asynchronous distinction is probably enough.

Our intention was to refer to coupling in which one executable calls individual components as runtime libraries as "online" coupling and coupling in which models are restarted with variables being exchanged through files as "offline" coupling. However, we realise that this distinction is not essential to our paper, and have removed mention of online and offline coupling. Instead we mention runtime and libraries where needed.
We also remove "partial" to avoid confusion (see also later comment and response below).

- L.102: there is a subsection 2.1.1 but no subsection 2.1.2. I suggest putting every- thing under 2.1 with no 2.1.1 subsection.

There is now also a Section 2.1.2.

- L.113: indicate that the user manual is provided on github or as supplementary file.

We provide the FISOC manual as a supplement as this is requested by GMD. However, the version of the manual in the repository is evolving while the attachment will become more out of date over time. For this reason we only add mention of the repository version in response to the reviewer.

- L.135-138: in which circumstances is it necessary to extrapolate? Is the ocean grid extends beyond the ice grid, I would expect the ice geometry (seen by the ocean) to be taken from observational data, not extrapolated from the ice model.

Given that ice geometry evolution is one of the main purposes of coupling ice sheet and ocean models, we do not envisage a situation in which using observational data for a part of the domain would be beneficial. In particular, as the ice model geometry evolves, a non-physical discontinuity between modelled and observed geometry is likely to occur. We do not intend the coupled system to be used in such a way and do not feel that discussing it would benefit the current paper.
Extrapolation in the current paper is due to the ocean model using a staggered grid which also includes ghost cells. In this situation the ocean domain extends beyond the ice domain by one and a half ocean grid cells. We've re-ordered the sentence to make it clearer to the reader that this grid stagger and ghost cells are the cause of the need to extrapolate.

- L.151: ". Cavity geometry. . ." add "seen by the ocean".
We've added this clarification (albeit with slightly different wording).

- L.161: It is unclear what is meant by "partial restart" here. In Favier et al. (2019), the ocean is restarted every coupling interval by conserving its velocity field (and all infor- mation on the current and previous time steps that is usually used to restart the ocean model), not only temperature and salinity as in De Rydt et al. (2016), so shouldn't it be called a full ocean restart. The

main difference between Favier et al. (2019) and ROMS/FVCOM here is that the ice geometry seen by the ocean evolves by step, i.e. every coupling interval rather than every ocean time step, and that there is an associated correction on barotropic velocities to cope with that.

Our use of the term "partial restart" was ambiguous, and it could also be that we were not up do date with how different groups are implementing their restarts. We've removed "partial".

- L.311-312 (or in section 2.6): please specify whether the 3-equation formulation is used with a constant exchange velocity. It could be worth mentioning that "rotation is disabled" to avoid asymmetric melt rates and seek a perfectly flat solution for the ice draft.

The exchange velocity is a function of friction velocity. We've now added this in the section on thermodynamics at the ice-ocean interface.
We have now stated in the experiment description section that rotation is disabled.

- L.320-323: if these details are given for FVCOM, I guess they should be provided for ROMS as well. Having said that, I am not sure these are useful in this paper, unless there are reasons to think that some schemes are less conservative than others.

We have added similar information for ROMS. We are not investigating the impact these schemes have on conservation, but they may have some relevance to melt rates. This is also not a focus for our study, but the information may be of some interest in this context.

- L.311 and 353-354: are these in-situ or potential temperatures?

This is potential temperature. We have clarified this in both places.

- section 4.1: it would be worth mentioning whether the ocean models make the Boussi- nesq approximation (which would probably mean that they are expected to conserve volume rather than mass). Is the ocean mass in Fig. 4 derived from the uniform sea water densities shown in Tab. 2 based on its volume?

Yes, the ocean models make the Boussinesq approximation.
Yes, the ocean mass in Figure 4 is derived from volume multiplied by the uniform density.
We now mention both of these things where Figure 4 is first referenced.

**2   Response to reviewer 2, Xylar Asay-Davis**

This paper describes a framework, FISOC, built on the Earth System Modeling Framework (ESMF), for coupling ice sheet and ocean components. Since ESMF is used for coupling in many Earth System Models (ESMs), the authors suggest that FISOC could provide an important stepping stone toward ice sheet-ocean coupling in an ESM. The paper describes the coupling infrastructure as well as the ice sheet and ocean components used for coupling verification. Then, the authors use two idealized test cases to demonstrate approximate conservation of mass and consistency between the grounding line as represented in each

component.

The paper is well written and well organized. The figures do a good job at illustrating the design concept of the framework and its flexibility in addressing the unique requirements of the components it currently supports. For the most part, I find the description of FISOC and the verification experiments appropriately detailed and easy to follow. There are a few areas where I think more clarification or detail would be useful, as detailed below in the specific comments, before the paper is ready to publish in GMD.

The authors wish to thank Xylar Asay-Davis for his positive comments, thorough review and useful suggestions. We have added the requested clarification and information, making for a more complete paper.

First, the text mentions briefly that FISOC currently uses "sequential coupling" (but that "concurrent parallelization" would presumably require minimal effort). However, the text does not provide sufficient detail on how sequential coupling is performed, in particular what the conceptual start and end time of each component's coupling interval is. Nor is there any description of how this might be different for concurrent parallelization. I think these are needed to better understand the coupling strategy and, in particular, the inconsistencies between the geometry as represented in each component.

We have added a subsection on sequential parallelism in section 2. This clarifies the sequential workflow with new text and a new figure (Figure 2 in the revised manuscript). Further speculation on concurrent parallelism is also added in this new subsection.
The results section for experiment VE1 now also has some additional text near the end which includes mention of the sequential lag.

Second, there is no discussion in the paper about how the ocean components ensure ocean connectivity (if at all) and how this may need to be accounted for in the coupling. Would the ocean components allow melting in "subglacial lakes" (as emerge in the experiments of De Rydt and Gudmundsson, 2016) that meet the flotation criterion but are not connected to the ocean? If, so, this could drive unrealistic ice-sheet dynamics. Would these subglacial lakes be considered part of the floating area in the ocean component? If not, would this lead to a significant discrepancy in the geometric representation (or at least accounting) between the two components?

Neither of the ocean components make any attempt to ensure connectivity. "subglacial lakes" are allowed to occur. We now mention this at the end of the "grounding line evolution" section.
The ice component uses it's own grounded mask to avoid application of melt rates to grounded ice. This was applied to all simulations presented here but we were remiss in not explicitly stating this in the paper. The implication is that any geometry change due to melt occurring in locations the ocean component considers floating but the ice component considers grounded is effectively removed from the coupled system. This doesn't happen in the current simulations in which the ocean grounding line retreat slightly lags the ice grounding line retreat, but will be considered carefully in future studies. In particular, we will aim to incoroporate ESMF conservative masked regridding in the future.

We now mention the application of the ice component grounded mask to prevent applying melting under the section on "Thermodynamics at the ice-ocean interface".

One should also consider the possibility that allowing the ocean model to represent subglacial lakes could drive MORE realistic dynamics, and we hope to consider this in future studies, in which a subglacial hydrology model will be incorporated in the coupled system.

Third, while emphasis is placed on conservation, the interpolation methods used in the VE1 experiment are not conservative and therefore would not be appropriate for flux fields (like the melt rate) in ESMs. Relatedly, the results from VE1 demonstrate approximate, but not machine-precision, conservation of mass. Could this be improved by using conservative interpolation and also accounting for the mass accumulated in the coupler during a coupling interval?

We have added relevant content in two places.

1. We now discuss briefly the potential applicability of ESMF conservative regridding methods in the regridding subsection of section 2.

2. We discuss the sources of error in mass conservation at the end of the results section "VE1: Floating adjustment"

More generally, we aim to place emphasis on presenting the coupling rather than conservation. One experiment is set up in such a way that conservation can be assessed. It is clear that the reviewer would like to see more discussion and more detailed work generally on this topic. We agree that it is important, but our current setup is not aimed at providing close to machine precision and is not aimed at quantifying causes of mass drift. the reviewer's comments have helped us to consider the various factors contributing to non-conservation and we will investigate this in the future at the level of detail that the reviewer would like to see. We intend to do this once we have a setup in which the ESMF conservative regridding methods can be used within FISOC. For the current study, we have added relevant information and discussion in a number of places, though we recognise that this is not at the level of detail that the reviewer would ideally like to see.

Fourth, I found the geometry and design choices of the VE2 experiment hard to follow. It would be helpful to have a figure showing the initial side-view (x-z) geometry for the experiment as well as a cross-section of the geometry at 25 years shown in x-y in Fig. 5. It would also be helpful to have a velocity plot for the ice-sheet component similar to that for the ocean components in Fig. 6. The behavior of VE2 strikes me as quite dissimilar to realistic ice sheet/ice shelf dynamics in that the thickest part of the ice shelf is in an ungrounded region and (as near as I can tell) ice seems to be flowing out of the "inflow" boundary at x = 0. It seems like some acknowledgement of the rather significant limitations of this experiment are needed somewhere in the discussion.

The experiments were intended to be the simplest configurations in which a 3D coupled ice sheet - ocean modelling system could be demonstrated. They were never intended to provide a directly relevant abstraction of a real world set up like the MISOMIP experiments (which to some extent resenbles the Pine Island Glacier, or an embayed marine system more generally). FISOC is intended

to be applied to Antarctic systems in the first instance, and so the experimental design needed to feature large ice shelves. The only real requirement we had of VE2 is that it features a large ice shelf and has an evolving grounded region. We have added a sentence about the design at the start of the experiment description section for VE2.
We have replaced the previous Figure 5 with a double plot showing centerline profiles (Figure 6 in the revised manuscript). This shows the geometry at the start and also distribution of the ice flow speed along the domain. This aims to address the reviewer's requests for a geometry cross section and ice flow plots. The ice flow speed at the inflow boundary is close to zero for much of the simulation but is never actually negative. We've added clarification of this just after the new figure is referenced in the text (VE2 results section).

As long as the authors make an effort to address these comments or explain their reasoning for not addressing them, I do not need to see a revised manuscript before publication.

**2.1 Specific Comments**

l. 53: ESMF does not need to be redefined, since it was already defined on l. 47-48

Removed, thanks.

l. 69-70: I think this is an important point that should be explored in a new subsection of section 2. Presumably, both components start at t=0. Which component runs first? Let's say it's the ocean. Once the ocean has finished a coupling interval, it has computed a melt rate. Is this averaged over the coupling interval or is the instantaneous value at the end of the coupling interval used? (This has important implications for how precise conservation of mass will be computed.) Presumably, the $dD/dt$ is initialized to zero in the ocean component, so this is clear for the first coupling interval, but I will come back to this. Then, let's say the ice-sheet component runs. It is able to apply the known melt rate for the coupling interval (10 days in VE1 and VE2), so there is no conceptual time lag here if the melt rate is a time average but there is one if it is an instantaneous value from $t = 10$ days. Based on the results of this coupling step, $dD/dt$ can now be computed and interpolated to the ocean grid or mesh. This $dD/dt$ is time-centered at $t = 5$ days, but will be applied over days 10 to 20 in the ocean component, so this is the source of a time lag that you discuss later.

If the components run in the opposite order, it is conceivable that the time lag could be placed on the melt rate instead of $dD/dt$. There is no explicit description of this, but I get the sense that this was not the choice that was made, since results from VE2 discuss a lag in D, not in melt rate.

If I am correct in assuming that the ice sheet component updates second, after the ocean, in a given coupling interval in the sequential scheme, this likely means you do not need to account for mass that conceptually accumulates in the coupler during a coupling interval. It seems important to me to discuss that "concurrent parallelization" will require a time lag in both $dD/dt$ and in

melt rate, since each component will be updated based on the state (or time average) at the end of the previous coupling interval. In this scenario, it would be important to keep track of the mass that conceptually accumulates in the coupler over a coupling interval over a time step. This is the approach used for fluxes between components in the Community Earth System Model (CESM) and Energy Exascal Earth System Model (E3SM), the ESMs I am most familiar with, and I think in other ESMs as well. Even in cases where components may run sequentially on the same processors, I do not think it is common to take advantage of this to remove the time lag in fluxes between components because of the conservation issues that could arise.

Again, I feel like some discussion of these nuances is missing from Sec. 2.

We have added a subsection on sequential parallelism in section 2 (subsection 2.1.1). This clarifies the sequential workflow with new text and a new figure (Figure 2 in the revised manuscript). It also mentions the lag related to our sequential coupling. We do not think that this lag imposes a mass conservation drift. Further speculation on concurrent parallelism is also added in this new subsection.

We have added a paragraph about time processing of variables to the section on "coupling timescales". This clarifies that ocean time averages are used in the current study.

We note that sequential and concurrent parallelism can both be run in such a way that the same lag is present in both components. Concurrent paralleism is a technical design choice affecting efficiency, but it does not offer any new options regarding variable lag (unless the coupling were to be implemented at a sub-timestep level, which is unlikely to occur between ice dynamic and ocean models). For this reason we do not discuss the lag in the context of sequential vs concurrent parallelism.

Given that, in the current study, the ice model timestep is equal to the coupling interval, and that the ocean model is also called for the same period (the ocean components themselves decide how many ocean timesteps are needed), the concept of mass (or any state property) "accumulating in the coupler" is not relevant to the current study. If, in the future, we call either component for a period shorter than the coupling interval then FISOC will need to handle cumulating variables. The code for this is in place but we do not feel that discussing in this paper is a useful direction. It will be introduced and discussed in the future as and when we need it.

l. 85-86: ESMF does not need to be reintroduced and the citations are not needed because the acronym is already defined on l. 47-48 and the citation are already covered on l. 69-70.

Removed, thanks.

l. 134: "All FISOC simulations to date have used a Cartesian coordinate system." Is Elmer/Ice capable of using a spherical coordinate system? The BISI-CLES and MALI models that I have worked with both work only on Cartesian (polar stereographic) meshes, which requires special care to ensure flux conservation but can be handled as long as the coupling infrastructure is aware of the discrepancy in areas between the component models.

*Standard Elmer/Ice options include Cartesian and cylindrical coords. Regridding between components running on Cartesian coordinate systems is surely the least problematic option. Possibly the reviewer refers to difficulties regridding between components using different projections or coordinate systems? But here, as noted, all components use Cartesian coordinates.*

l. 155: "FISOC assumes that time-step sizes are not adaptive." This seems quite restrictive to me and potentially unnecessary. It seems like this could use some discussion. I have worked with the BISICLES and Parallel Ocean Program coupled model called POPSIClLES. In that model, we had a different coupling strategy and we always ran with concurrent parallelism over a coupling interval. We found many cases doing more realistic simulations where it was highly beneficial that BISICLES (which can perform adaptive mesh refinement) could refine its time step to handle a particularly tricky geometric configuration that might emerge spontaneously. We simply required that BISICLES perform a time step that exactly reached the coupling end time as the last step in a coupling interval. It seems like this strategy would also be compatible with FISOC, and therefore the requirement that the coupling interval is equal to the ice-sheet time step is unnecessarily restrictive. If there are important reasons for the restriction, it would be helpful if they are clarified. If this is not a strict requirement but rather has been the convention in simulations to date, this should be discussed.

*We've made modifications to this paragraph to clarify that these restrictions do not always need to be imposed.*

Eq. (1): I think some more nuanced discussion is needed about the time-centering of $dD/dt$ (which is at $t - 1/2$ Delta t) and when the $dD/dt$ is actually applied in the ocean component (centered at $t + 1/2$ Delta t). You have $dD/dt$ subscripted at time t but I do not think that is correct in either component.

*This equation is correct if time t corresponds to the end of an ice component timestep. We have clarified this in the text. the reviewer's comments here essentially relate to the lag inherent in our approach to sequential coupling, which is now discussed in the new subsection 2.1.1 on "sequential parallelism".*

l. 180: It is important to clarify that D is positive up. This might seem obvious from an ice-sheet modeling perspective, where D being positive down would not be an obvious choice since it can take either positive or negative values. But D in the ocean is often used for "depth" and is almost universally a positive quantity, so the choice of variable names and the sign convention are not intuitive for ocean modelers. There are also some later equations where I think the sign of D is not correct (as I will point out), leading to further confusion about the sign convention. A lot of confusion in the paper might be spared by renaming this variable "z_d" to go with "z_b" for the bedrock elevation/bathymetry.

*Thanks for spotting this mistake. We assumed D to be positive down to start with though we didn't state this. Then later we state that D is positive upward.*
*We have replaced D with z_d as suggested, and this is positive up everywhere.*

l. 183-184: "but has the potential for the ice and ocean representations to diverge over time as a result of regridding artefacts": Isn't some part of the divergence in time likely to come from the fact that there is a time lag between dD/dt from the ice component and as applied in the ocean component?

We are not convinced that a time lag can cause divergence in geometry over time. The integrated geometry change seen by the ocean after $n + 1$ ice timesteps would be (excluding regridding artefacts) the same as the integrated change given by the ice component after n timesteps. This is just a lag, not a source of divergence.

l. 213: "...FISOC can pass the temperature gradient from the ice component directly to the ocean component." I think this requires some discussion. The temperature at the ice-ocean interface is computed on the ocean time step and could potentially have significant temporal variability within a coupling interval (e.g. because of ocean eddies). Would the ice sheet get the time-average of this field as one of the coupling fields, and use this to compute the temperature gradient? If so, this would result in the temperature gradient going back to the ocean having a time lag of 2 coupling intervals in the temperature at the ice-ocean interface. Maybe this doesn't matter.

The temperature gradient in the ice should change pretty slowly compared to a coupling interval on the order of days. But this discussion is very speculative since we do not carry out thermodynamic coupling to the ice component in the current study. We prefer to focus discussion on the currently presented features of FISOC and leave detailed consideration of thermodynamic coupling to the future.

An alternative approach would be to pass the temperature in the bottom ice layer (and the ice thickness) and allow the ocean model to compute the temperature gradient on its time step. The differences between these approaches is likely only to matter for particularly long coupling intervals but it might still be worthy of some though and some discussion. The choice is not entirely obvious, at least not to me.

The temperature gradient is very unlikely to be linear through the ice shelf. The gradient is likely to be much steeper near the lower surface (except in significant freeze on zones where the opposite would occur). Again, this could be a lengthy discussion that doesn't need to occur here.

Eq. (3): I believe the RHS of this equation needs a negative sign if D is positive up. Otherwise, the pressure would be negative.

Eq. (4): There is a sign problem with this equation, too. I am pretty sure it is that the whole RHS needs a negative sign again. If drho_o/dz were 0, you expect a positive pressure for a negative D. Later, drho_o/dz is given as a positive number, which is not physically reasonable if z is positive up. Density should decrease toward the ocean surface. But if that term is positive, pressure should increase because of increasing density at depths, so the term -0.5 drho_o/dz D_[O] is positive for negative D_[O], as it should be. If drho_o/dz is changed to be negative (as I think it should be), the sign of this term would also need to be flipped. In any case, there's something to be fixed here. The confusion may arise from an ocean component that uses a positive-down definitely of z, but I

think the paper needs to pick positive-up for everything and stick with it.

l. 245: "z_b is the bottom boundary depth (bathymetry, aka bedrock depth)": Most times the term "depth" is used in ocean modeling, there is an implication that it is positive-down. The fact that the variable is called "z_b" might tend to counteract that but I would state explicitly that it is positive-up.l.

In response to the last three reviewer comments: After changing ice draft from $D$ to $z_d$ we have corrected all instances of an incorrect sign.

246: "D_crit is a critical water column thickness (or depth)": This one is strictly positive, and is unrelated to D, which I find pretty confusing. Again, renaming D to z_d would do a lot to help with this. By the way, I don't see how the "or depth" bit applies at all in this context.

We removed "or depth".

Eqs. (6) and (7): I'm having trouble following these. An illustration would help a lot, but some text carefully defining the variables involved might do the trick.

The original definition was "$\eta$ is the free surface variable". In the new context, it's pretty hard to understand what $\eta$ is. The best way I can understand it is that $\eta + D_{-}[O]$ is the ocean's representation of the location of the ice-ocean interface, which is allowed to move up and down because of changes in ocean dynamic pressure. Maybe some explanation along these lines would be helpful.

the reviewer is essentially correct in his interpretation of $\eta$ (now renamed to $\zeta$ as it is usually termed in the ROMS community).
We've added some explanation along these lines about both the ice draft and $\zeta$ at the start of the section on "Handling cavity evolution".

As far as I can tell, Eq. (7) is equivalent to Eq. (6) except that D_crit is now a minimum ice-sheet thickness below flotation rather than a minimum ocean-column thickness? This is confusing and needs some explanation as to what exactly it means and why ROMS chose this definition instead of the simpler one from FVCOM. It is confusing to use the same name for variables with distinct meanings in the two models. Also, shouldn't a slightly different D_crit be used for ROMS to get the same ocean-column thickness (assuming this is desired)?

The definition of D_crit has not changed. It has the same meaning in both models.
We agree that interpreting the ROMS wet/dry equation is not intuitively obvious. We've added a couple of lines immediately after the equation to describe conceptually the ROMS wet/dry criterion.

l. 257: "as described in Section 2.8": The current section is 2.8, so this must be a mistake. Maybe the reference is supposed to be to Sec. 2.5?

Corrected, thanks.

l. 272-274: "The coupling is purely geometric in that the ocean component passes an ice shelf basal melt rate to the ice component and the ice component passes a rate of change of ice draft to the ocean component." This may be a nuance of interpretation but I do not think of the mass flux in the form of a melt rate as being a geometric quantity, so I would disagree that the coupling is purely geometric.

we modified this wording to "The coupling centers on the evolution of ice geometry"

l. 280-282: I think it would be helpful to have an explanation for why the FVCOM simulation required a domain of a slightly different size. It is not clear if the sizes given in Table 2 are for both components or just the ocean component (in which case a row is needed for the ice component).

We added a line later in the same paragraph to clarify that the ice domain matches whichever ocean component it is coupled to, and to explain why the FVCOM domain has slightly different size.

l. 295: "$\rho\_or = 1027 kg m^{-3}$": You give a slightly different value for FVCOM in Table 2 but this difference is not addressed here or anywhere else. Why the difference?

Sorry, the FVCOM reference density is not actually used anywhere in the current study! We've removed it from the table.

l. 297-299 and Eq (9): You gave a definition of the pressure at the interface in Sec. 2.7 already, and it was different from this for FVCOM. Presumably this redundant definition is not needed.

The pressure is given here in the context of the ice sheet boundary condition. It matches the earlier equation for ROMS but has a small discrepancy when coupled to FVCOM. We have changed this text so that it now refers directly to the earlier equation instead of repeating it.

l. 303: I think "zero net accumulation" is a slightly confusing phrase here. I assume the idea is that ice sheet models typically have a field of net accumulation (called a) and that this is zero everywhere in this case. But it lends itself to the misunderstanding that there is accumulation but that the net effect is zero (e.g. averaged in time, space or both). Could this be simplified but just removing the word "net"?

Yes, we removed "net".

l. 315 "ROMS specific details." Nothing is given about vertical mixing or eddy parameterizations, whereas these details are given for FVCOM. No details are given about how the three-equation parameterization is handled in either ocean component. For example, where are "far-field" temperature and salinity sampled? What parameters are used? (Are they the same for both models?) Which equation of state and equation for the freezing point is used in each.

We have now added ROMS specific details that were previously given only for FVCOM.
We don't aim to reproduce the full description of the ocean component implementations of cavity physics as these are already described in existing papers. We have now given slightly more information, and also repeated the relevant references, in the section describing thermodynamics at the ocean interface.

l. 317: "FVCOM specific details." In addition to the above, no details are given about time stepping for FVCOM as they are for ROMS.

We have added the time-step information under the FVCOM specific details.

Eqs. (10) and (11): As I stated in my general comments, I think a figure is needed to help better understand this experiment. A starting point would

be a side-view (x-z) figure showing the initial ice-sheet, ice-shelf and ocean configuration as given by these equations. Also, since rho_or is slightly different for the 2 ocean components, is (11) accurate and H is therefore slightly different for the two but D is the same?

We have added a new Figure (Figure 6 in the revised manuscript) showing centreline geometry at the start and after 25 years.
$\rho_{or}$ is not actually used in FVCOM, and so we have removed it from Table 2.

l. 338: "No restrictions to ice flow are imposed at the upstream and down stream boundaries". I have several difficulties here. First, some more explanation is needed about what "no restrictions" really means. Presumably, this means that ice is free to flow out of the boundaries. I do not see how ice can flow in through these boundaries if there is "no restriction". Is it necessary to calculate stresses at the boundaries and, if so, how is this handled (in particular driving stress)? Why was an open boundary condition like this chosen at x=0? A more typical setup would place a solid boundary here so it acts as something of an ice divide. This would also make the direction of ice flow a lot less ambiguous. That brings me to the second point, which I will discuss more below. The ice flow field is not discussed but I get the impression based on the thickness evolution that flow is happening out of both the x=0 and x=100 km (or 99 km) boundaries, so that the "upstream" and "downstream" directions aren't well defined in this problem.

The "no restrictions" was sloppy wording on our part. The inflow and outflow boundaries have appropriate external pressures prescribed. We have revised the experiment description to convey this.
In effect this means that ice is allowed to flow either out from or in through both the "inflow" and "outflow" boundaries. In practice ice only ever flows in through the "inflow" boundary and out through the "outflow" boundary, though the velocity is close to zero at the "inflow" boundary. We've added a new Figure (Figure 6 in the revised manuscript) showing profiles at the start and after 25 years as well as ice flow speed.
This experiment is not intended to represent an ice catchment extending to the ice divide. It simply aims to provide a large shelf and a grounded region that evolves. It would have been possible to impose a no-flow inland boundary, and we do not think this would significantly alter the applicability of the experiment to demonstrating coupling.

l. 364-366: I found this paragraph redundant to the paragraph on l. 268-270 and subsequent text. I realize it is nice to summarize things again from previous sections but this seems too repetitive to me.

Yes, this is redundant. We removed these lines.

l. 372-373: I don't think "along the domain" and "cross-domain" are well defined directions because they take the perspective of the ice flow in a context where ocean circulation is being discussed. I would just call these the x and y directions.l.

382-383: "The net mass change of the coupled system is more than an order of magnitude smaller than the mass change of the individual components for both experiments VE1_ER and VE1_EF." I think this needs considerably more

discussion. For ESMs, anything less than machine-precision conservation is not considered acceptable and is one of the most important mechanisms for diagnosing model inconsistencies. To accomplish this, conservative regridding is always used for flux fields (the melt rate in the case of FISOC, and the heat flux in the future). Ice sheet-ocean modeling requires that special care must be taken to distribute that flux field to the ice-sheet component because melt should not get distributed to grounded cells by mistake but it also should not be lost in the regridding process because this would affect conservation. This issue is exacerbated by inconsistent representation of the grounding line between components. Is this taken into account in FISOC? If so, please discuss. If not, please discuss this as a potential issue for future consideration. Aside from interpolation, conservation of mass may be inexact in FISOC because of the lag between when melt rates are computed in the ocean component and when they are applied in the ice component. I convinced myself when I was discussing the staggered coupling approach above that this is likely not the case in the current approach but it would be in an approach with concurrent parallelism. Even so, it would be important to diagnose that total mass going into the coupler is exactly equal to total mass coming out of the coupler after each coupling interval (i.e. after both components have run) or that the difference between these two is computed and stored within the coupler to be distributed appropriately at the next coupling interval. Overall, I would like to see some discussion about why conservation of mass in FISOC is good but not machine-precision good.

Much of this overlaps with the reviewer's earlier comments and some requested discussion has been added (see above for details). It is true that component's mask discrepancies has the potential to impact on conservation, and a comment on this has been added to the section on "Thermodynamics at the ice-ocean interface" where we first talk about the ocean's basal melt rate which is passed to the ice component.

l. 386-387: "While the initial slope of the lower surface of the ice shelf is the same in both VE1 and VE2": I misunderstood this the first time I read it to be saying that the D's for VE1 and VE2 were the same. They differ by 20 m but the slope is the same. I guess it's fine as it is but I wanted to let you know about the confusion, in case you want to do anything about it.

We discussed this and are currently happy that the meaning is sufficiently clear.

l. 387: "the open inflow and outflow boundaries": I remain confused about the open "inflow" boundary at x = 0. Is it really inflow? If so, how does open inflow work?

Yes, it is inflow, though very slow. Pressures are prescribed. See our response above to where the reviewer's concern is first raised for a more complete response.

Fig. 5: First, as I stated in my general comments and as I think you are fully aware, this is an odd ice-shelf geometry. It is also not very intuitive to see thickness plotted as an x-y field, at least not for me. It would be more helpful in my opinion to have a more 3D plot similar to Fig. 3. It would also be really helpful to have a vector field for the ice like the one for the ocean velocity in

Fig 6, especially because I want to see how much the weird geometry is due to outflow (instead of inflow) at x = 0.

We've replaced this Figure with a Figure showing centerline profiles at t=0 and after 25 years. (FIgure 6 in the revised manuscript) This gives the reader a more direct visualisation of the initial and evolving ice geometry. We've now also included ice flow speed in the new Figure 6. We show this as a coloured field and not as suggested by a vector field because the flow direction is entirely dominated by the x component (the arrows all point the same way).
As mentioned in an earlier response, there isn't any flow out of the domain at x=0, though we appreciate that it would be confusing to the reader to think that this could be the case. The new Figure should make it clear that there is no outflow at x=0.
We would add the comment though that it is not necessary for this domain to resemble a real glacier. It would not be a problem for there to be outflow at x=0, so long as the grounding line evolves and can be analysed.

Fig. 6: Are these fields interpolated to a common, regular grid? They look like they might be and, if so, this should be mentioned.

Yes, FVCOM was regridded to the ROMS grid, then both subsampled at 2km. We've added this information to the figure caption.

l.415-421: It may be worth remarking that the difference in grounded area does not increase with time even with the Rate method, at least in this case.

We added one sentence about this.

**2.2   Typographical and grammatical corrections**

We have fixed all the typos listed below. Separate author responses to each are not needed here.

l. 7: "these mechanisms" missing a space

l. 8: a comma is needed between "this" and "ocean"

l. 22: "(MISI) (Mercer, 1978; Schoof, 2007; Robel et al., 2019)": I would combine these parentheses as you have done elsewhere: "(MISI; Mercer, 1978; Schoof, 2007; Robel et al., 2019)"

l. 38: Similar to above: "(ISOMIP+; Asay-Davis et al., 2016)"

l. 46: commas are needed: "Here, we present a new, flexible..."

l. 46: for consistency, a semicolon is needed instead of a comma: "(FISOC; Section 2)"

l. 47-48: "Earth System Modeling Framework" is a proper name so I think it needs to be spelled with the American version of "Modeling" that is used on their website: https://www.earthsystemcog.org/projects/esmf/

l. 53-54: "(Hill et al. (2004); Collins et al. (2005))" should be "(Hill et al. 2004; Collins et al. 2005)" without the nested parentheses.

l. 105: a comma is needed between "versa)" and "all"

l. 117: a comma is needed between "cases" and "a non-standard"

l. 120: "Regional Ocean Modeling System" is also spelled with the American spelling of "Modeling" in the documentation I could find: https://www.myroms.org/

l. 120: There should not be nested parentheses: "(ROMS; Shchepetkin and McWilliams 2005)"

l. 120-121: For consistency, this should be "terrain-following, sigma-coordinate" (with a comma and a second hyphen). In general hyphenation is used a lot more sparsely in this writing than I would use it but I fully acknowledge that that is a stylistic choice.

l. 123: Remove the nested parentheses: "(FVCOM, Chen et al. 2003)

l. 136: a comma is needed between "extrapolation" and "which"

l. 142: "time step" is not typically hyphenated but this may be a stylistic choice.

l. 174: a comma is needed between "used" and "this"

l. 176: a comma is needed between "large" and "occasional"

l. 194: a comma is needed between "case" and "the user"

l. 209: "ocean model ice shelf cavity shape" is quite a long compound noun...

l. 228: "kg m" needs a space or half-space

l. 247: a comma is needed between "Thus" and "cells"

Eq (7): Please remove the asterisk as a multiplication symbol. It is not needed and is not considered a valid multiplication symbol (outside of code).

l. 258: a comma is needed between "Study" and "dD/dt"

l. 275: I've left most of the hyphenation choices alone but I feel pretty strongly that "uniform-thickness" should be hyphenated.

l. 286: a comma is needed between "system" and "we"

l. 287: a comma is needed between "Therefore" and "the"

l. 288: a comma is needed between "corners" and "where"

Eq (9): should end in a comma, not a period.

l. 309: a comma is needed after "experiment" at the end of the line

l. 326-327: commas are needed after "VE1_ER" and "VE1_EF"

l. 338: "down stream" should be "downstream"

l. 360: a comma is needed between "interval" and "this"

l. 368-369: a comma is needed between "days)" and "the" and again between "years)" and "the"

l. 378: a comma is needed between "melting" and "the"

l. 380: a comma is needed between "system" and "the"

l. 388: commas should be removed from "...component and the relatively shallower ice in the grounded region both..."

l. 397: a comma is needed between "melting" and "as"

l. 400: a comma is needed between "2.8" and "the"